# Cortical flow aligns actin filaments to form a furrow

Anne-Cecile Reymann[1,2], Fabio Staniscia[3†], Anna Erzberger[3†], Guillaume Salbreux[3,4]*, Stephan W Grill[1,2,3]*

[1]Biotechnology Center, Technische Universität Dresden, Dresden, Germany; [2]Max Planck Institute of Molecular Cell Biology and Genetics, Dresden, Germany; [3]Max Planck Institute for the Physics of Complex Systems, Dresden, Germany; [4]The Francis Crick Institute, London, United Kingdom

**Abstract** Cytokinesis in eukaryotic cells is often accompanied by actomyosin cortical flow. Over 30 years ago, Borisy and White proposed that cortical flow converging upon the cell equator compresses the actomyosin network to mechanically align actin filaments. However, actin filaments also align via search-and-capture, and to what extent compression by flow or active alignment drive furrow formation remains unclear. Here, we quantify the dynamical organization of actin filaments at the onset of ring assembly in the *C. elegans* zygote, and provide a framework for determining emergent actomyosin material parameters by the use of active nematic gel theory. We characterize flow-alignment coupling, and verify at a quantitative level that compression by flow drives ring formation. Finally, we find that active alignment enhances but is not required for ring formation. Our work characterizes the physical mechanisms of actomyosin ring formation and highlights the role of flow as a central organizer of actomyosin network architecture.

*For correspondence: guillaume.salbreux@crick.ac.uk (GS); stephan.grill@biotec.tu-dresden.de (SWG)

†These authors contributed equally to this work

Competing interests: The authors declare that no competing interests exist.

## Introduction

Cytokinesis begins in late anaphase, triggered by regulatory pathways with feedback from the mitotic spindle that ensure appropriate positioning of the molecular machinery (reviewed in *Fededa and Gerlich, 2012*; *Green et al., 2012*). As a consequence, the small regulatory GTPase RhoA (*Piekny et al., 2005*; *Tse et al., 2012*) and the molecular motor myosin (*Shelton et al., 1999*; *Werner and Glotzer, 2008*) are enriched in an equatorial band (*Piekny et al., 2005*; *Werner and Glotzer, 2008*). Myosin can directly organize the actin network (*Miller et al., 2012*; *Soares e Silva et al., 2011*; *Vavylonis et al., 2008*), and if myosin-based active alignment (e.g. via search-and-capture [*Vavylonis et al., 2008*]) or flow-based compression as proposed by White and Borisy (*White and Borisy, 1983*; *Rappaport, 1996*) drive furrow formation and cytokinesis in eukaryotes remains unclear (*Green et al., 2012*; *Bray and White, 1988*; *Levayer and Lecuit, 2012*; *Mendes Pinto et al., 2013*; *Cao and Wang, 1990*; *Murthy and Wadsworth, 2005*; *Zhou and Wang, 2008*).

## Results and discussion

We set out to address this question in the one cell embryo of the nematode *C. elegans* which forms two constricting ingressions, first a pseudocleavage furrow during polarity establishment, and then a cytokinetic furrow during cytokinesis (*Figure 1—figure supplement 1a*, *Video 1*) (*Munro and Bowerman, 2009*; *Rose et al., 1995*). Notably, the pseudocleavage furrow is linked to cortical flow (*Mayer et al., 2010*; *Munro et al., 2004*) and lacks regulatory control from the mitotic spindle (*Werner and Glotzer, 2008*). We first characterized the actomyosin cortical network during

**eLife digest** Just under the surface of every animal cell, a thin and dynamic network of filaments called the cell cortex acts as a scaffold and determines the cell's shape. When the cell divides, this material re-organizes to make a ring of filaments – known as the cytokinetic ring – across the middle of the cell. This ring then constricts to split the cell into two separate daughter cells. The filaments are guided to form the ring by specific proteins around the middle of the cell. A process called cortical flow – the mechanical compression of filaments towards the middle – also influences the shape of the ring. However, it is not clear to what degree cortical flow actually helps the ring to form.

A tiny worm called *Caenorhabditis elegans* is often used to study how animal cells divide and grow. When the *C. elegans* embryo is made of just a single cell, two rings of filaments form consecutively as this cell prepares to divide. Reymann et al. used microscopy to investigate how filaments are arranged in *C. elegans* embryos as the rings assemble. The experiments showed that filaments are arranged into rings in locations where the filaments are being mechanically compressed by cortical flow. The first ring forms and partially constricts, and then relaxes once the cell is polarized; that is, once the cell has developed two distinct ends. A second ring then forms during cytokinesis and constricts to divide the cell into two.

To understand the physical changes occurring, Reymann et al. compared the experimental data with a mathematical model of the cortical network. This model assumed that the cortical network acts as a thin film in which the orientation of the filaments is coupled to the flow of the fluid. Reymann et al. used this model to demonstrate that the observed arrangements of the filaments in both rings can be explained by cortical flow.

Together, the findings of Reymann et al. highlight the central role that cortical flow plays in organizing rings of filaments in *C. elegans*. Future studies will explore whether cortical flow is linked to other mechanisms that affect the formation of the cytokinetic ring.

pseudocleavage and cytokinesis. For this, we generated a Lifeact::mKate2 fluorescent line of *C. elegans* by coupling the Lifeact peptide to the far-red fluorophore mKate2 that was codon optimized for *C. elegans* (*Redemann et al., 2011*) (see Materials and methods and *Video 2*). This allows us to quantify cortical actin filament orientation in space and time, as well as cortical flow velocity fields by Particle Image Velocimetry (PIV, *Figures 1* and *2*, *Figure 2—figure supplement 1*) (*Mayer et al., 2010*). We find that actin filaments change from a randomly oriented and isotropic gel before cortical flows initiate, into a more circumferentially aligned organization at the future site of furrow ingression during both pseudocleavage and cytokinesis (*Figure 1b*, *Figure 1—figure supplement 1* and *Videos 1* and *3*). We studied how pseudocleavage and cytokinesis differ in the spatial distribution of molecular regulation and contractility within the equatorial region (*Figure 1*). Non-muscle myosin II (NMY-2) is the essential molecular motor that drives both cortical flow and

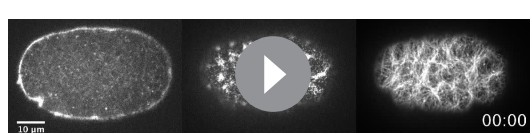

**Video 1.** Actomyosin gel dynamics in the *C. elegans* zygote. Cortical and medial planes of an embryo expressing both Lifeact::mKate2 and endogenous NMY-2::GFP. Left panel, medial plane Lifeact:mKate2, center, cortical NMY-2::GFP, right panel, cortical Lifeact:mKate2 (min:s).

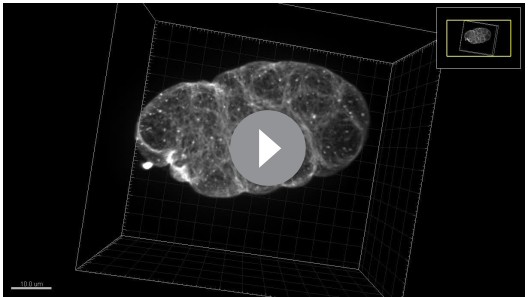

**Video 2.** Three-dimensional reconstruction of an early one-cell embryo of *C. elegans* expressing Lifeact:: mKate2. The back of the embryo is dimmer due to imaging.

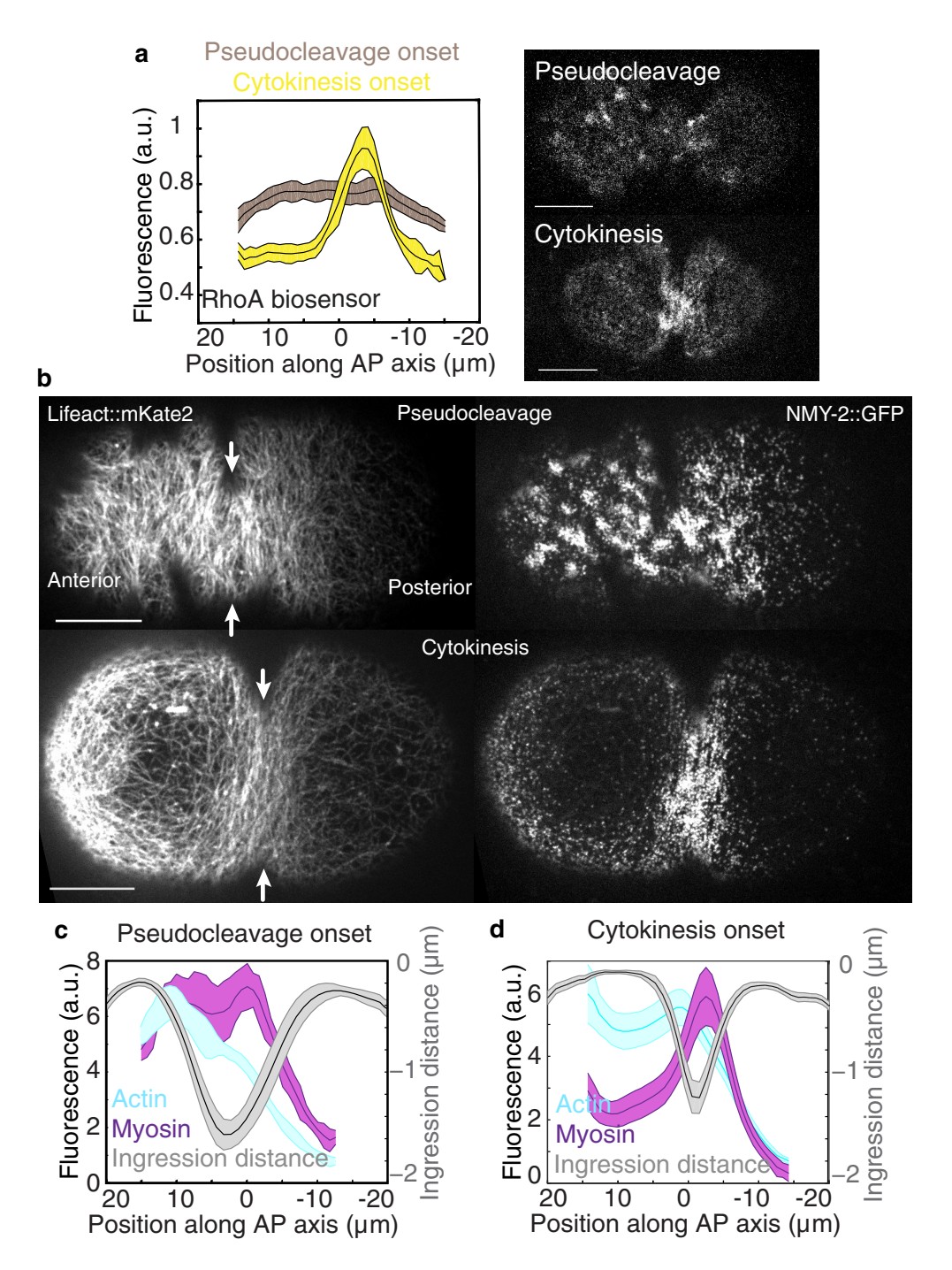

**Figure 1.** Flow and ingression during pseudocleavage and cytokinesis. (a) Average active RhoA biosensor intensity profiles along the AP axis (0 represents the embryo center; N = 14 embryos for pseudocleavage and N = 7 for cytokinesis; errors represent the standard error of the mean). (b) Actomyosin cortical organization at the onset of pseudocleavage and cytokinesis in embryos expressing both Lifeact::mKate2 and NMY-2::GFP. (c–d) Average actin, myosin and ingression profiles as a function of position along the AP axis (N = 10 embryos for pseudocleavage and N = 22 for cytokinesis; errors represent the standard error of the mean). Scale bars, 10 μm. The following figure supplement is available for *Figure 1*:

The following figure supplement is available for figure 1:

**Figure supplement 1.** Details of actin organization during the polarization flow phase.

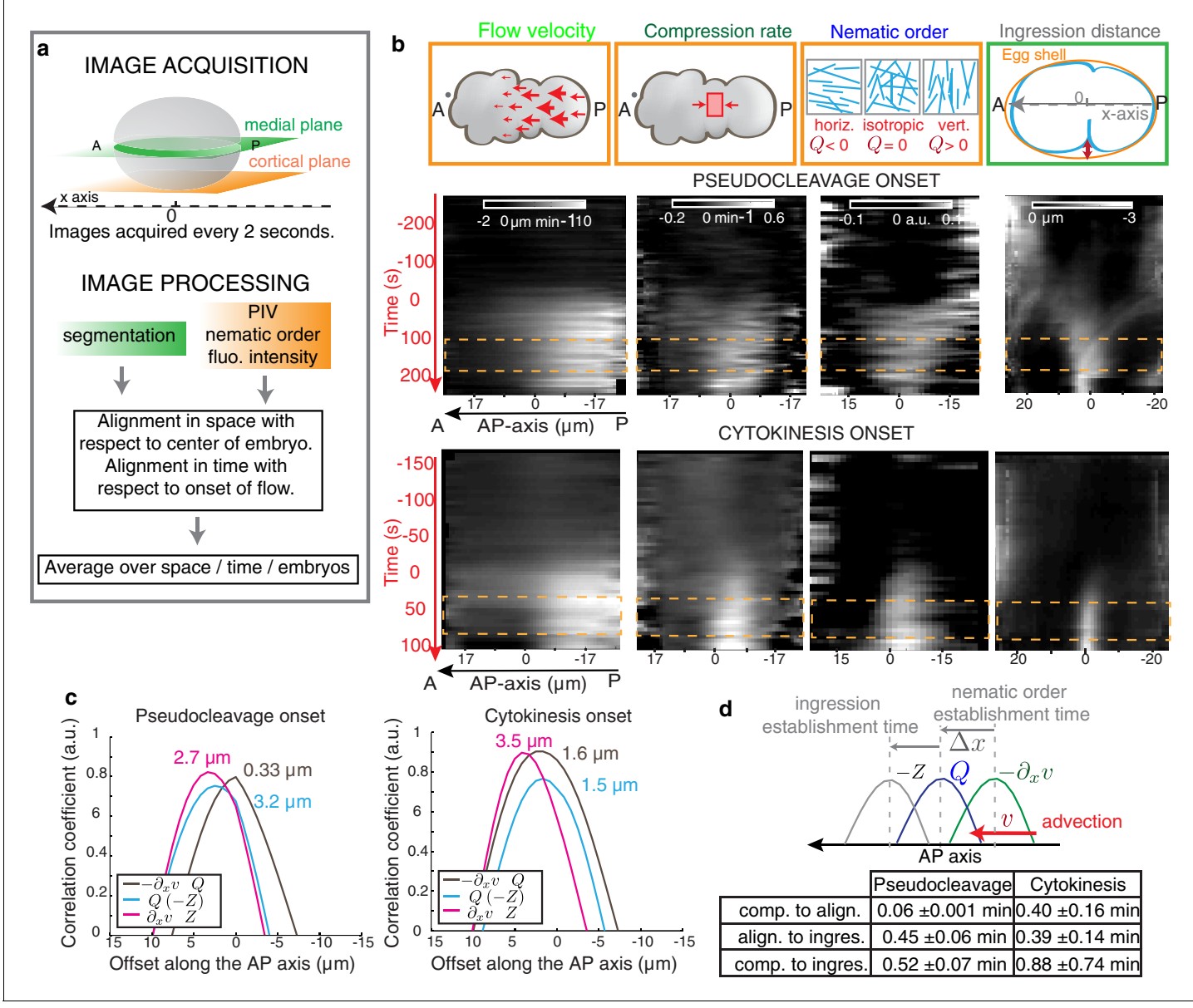

**Figure 2.** Cortical flows compress the gel to generate alignment. (a) Filament orientation is quantified by a nematic order parameter; flow fields are obtained from Particle Image Velocimetry (PIV) analysis. (b) Gel flow, compression rate, filament orientation and cell ingression fields. Top row, schematic representations. Middle row, kymographs of the time evolution of the profiles along the AP axis of flow velocity, compression rate, nematic order parameter and outer ingression distance for pseudocleavage (N = 10). Bottom row, cytokinesis (N = 12). Orange dashed lines indicate the times where all four fields are stationary. Flow, compression and nematic order were determined from bottom-section and ingression from mid-section views of Lifeact::mKate2 labelled embryos (see Materials and methods and Appendix). (c) Spatial correlation between compression, alignment and ingression (see Materials and methods). Peak value positions (estimated by a Gaussian fitting of the 1D correlation curve) are indicated. See *Figure 2—figure supplement 2* for spatiotemporal correlation functions. (d) Table of the temporal delays obtained from the correlation peak value positions using the mean values of the velocity in the 10 µm central region (mean values for pseudocleavage: $v$ = 6.16 ±1.23 µm/min; for cytokinesis: $v$ = 4.01±1.92 µm/min; mean ± SD).

The following figure supplements are available for figure 2:

**Figure supplement 1.** Quantification methods developed.

**Figure supplement 2.** Spatiotemporal correlations.

**Figure supplement 3.** Nematic order parameter quantification of artificial images with and without directional bias.
*Figure 2 continued on next page*

*Figure 2 continued*

**Figure supplement 4.** Impact of image quality on the nematic order parameter quantification.

**Figure supplement 5.** Comparison of different actin labeling strains and actin-binding protein localization.

cytokinesis (*Shelton et al., 1999*; *Guo and Kemphues, 1996*), whereas the small GTPase RhoA (*Piekny et al., 2005*; *Tse et al., 2012*), when active, is responsible for its local activation. At the onset of cytokinesis, active RhoA and NMY-2 locally increase at the position of the contractile ring, while for pseudocleavage we observed no such increase (*Figure 1*). Instead, the equatorial region in pseudocleavage corresponds to a transition zone with a gradual decrease of RhoA and myosin between the anterior region of high and the posterior region of low contractility (*Figure 1*) (*Mayer et al., 2010*). The actin nucleator formin CYK-1 is downstream of RhoA (*Piekny et al., 2005*) and follows a similar pattern of localization, and actin bundling via anillin (*Piekny and Glotzer, 2008*) or plastin also does not appear to be increased in the equatorial region during pseudocleavage (*Figure 2—figure supplement 5e–f*). To conclude, the pseudocleavage furrow ingresses with a circumferential alignment of filaments in the gel, but without a local zone of RhoA activation and the corresponding local increase in myosin and formin density. Since search-and-capture like mechanisms require a localized band of myosin and actin nucleators at the equator to drive myosin-based active alignment (*Vavylonis et al., 2008*), this suggests that actin filaments in pseudocleavage align via flow-based compression and not via active alignment.

We next sought to test if flow-based compression drives actin filament alignment. To this end, we developed tools for the spatiotemporal quantification of compression by flow, of filament orientation, and of cortex ingression distance within the gel both at the onset of pseudocleavage and cytokinesis (see Appendix as well as *Figure 2a* and *Figure 2—figure supplement 1*). We find that the flow compression rate along the antero-posterior direction (AP axis or *x* axis), given by the spatial gradient of the velocity field $-\partial_x v$ (*Figure 2b*), increases with time during the very early stages of pseudocleavage. The compression rate peaks in the central region of the embryo, with a maximum of ~0.4 min$^{-1}$. Notably, this peak is stable for several minutes until pseudocleavage flows cease entirely (*Figure 2b*). For cytokinesis, we find that the early flow also proceeds in a unidirectional manner from the posterior toward the anterior (*Figure 2b*). Similar to pseudocleavage, the compression rate increases with time over the very early stages of cytokinesis, with the compression profile peaking at the center of the embryo with a maximum rate of ~0.7 min$^{-1}$ (*Figure 2b*). We characterized the average actin filaments orientation by a nematic order tensor **Q** (*Figure 2—figure supplement 1a*). This analysis relies on the fact that the intensity of the Fourier transformed image encodes geometric characteristics such as its main orientation pattern (see Appendix). We find that vertical (orthogonal to the direction of flow) alignment appears coincidental in space and time with the local increase of the compression rate (*Figure 2b*, compare column 2 and 3). Notably, changing the direction of flow also changes the direction of alignment: off-site sperm entry leads to flows along the short axis of the egg (*Goldstein et al., 1993*) and actin filaments still align in the direction determined by compression (*Figure 1—figure supplement 1d*, *Video 4* and Appendix). Finally, the ingressing furrow forms in the region where filaments are aligned (*Figure 2b*, column 4). Our quantifications reveal remarkable similarities between the compression rate fields, the pattern of alignment, and the ingression distance between pseudocleavage and cytokinesis (*Figure 2b*), suggesting common mechanisms. To conclude, actin filament alignment arises at locations of significant compression by flow.

If flow-based compression aligns actin filaments for forming an ingression, we would expect compression to precede or to be concomitant with alignment and ingression. The 1D crosscorrelations between compression, filament alignment and ingression distance, calculated for the time that flows are essentially stationary (*Figure 2b* and *Figure 2—figure supplement 1*), show that for pseudocleavage both compression rate and alignment peak at about the same location, while the ingression field peaks ~ 3 μm further to the anterior (*Figure 2c* and *Figure 2—figure supplement 2* for the spatiotemporal crosscorrelations). The situation is similar at cytokinesis, except that the compression

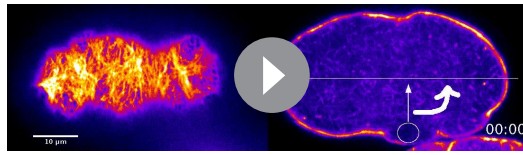

**Video 3.** Cortical dynamics at flow initiation. Left panel, cortical NMY-2::GFP, center cortical Lifeact:mKate2 and a zoom is shown in the right panel (min:s).

**Video 4.** Cortical dynamics of an embryo in which flow was initiated at the side (bottom) and far from the posterior pole of the embryo. Actin filament alignment follows flow and compression, thus transitions from a horizontal to a vertical direction (min:s).

field peaks furthest to the posterior. We next translated the characteristic distances between peaks in stationary flow to temporal delays between events (see Materials and methods for further explanation). We find that for pseudocleavage, filament alignment is essentially concomitant with compression, while for cytokinesis compression precedes alignment by 0.40 ± 0.16 min (*Figure 2d*). Furthermore, ingression arises approximately 0.5 min after compression for pseudo-cleavage and 0.9 min after compression for cytokinesis. To conclude, filament alignment appears to rapidly follow compression while ingression arises with a delay.

We next sought to provide a physical explanation for flow-based alignment (*Salbreux et al., 2009*). The cell cortex in this and other systems can be thought of as a gel that forms a thin layer underneath the plasma membrane. Recent advances in physical theory show that this gel is active and generates the forces that drive cell shape changes (*Salbreux et al., 2009*; *Gowrishankar, 2012*; *Kruse et al., 2005*; *Turlier et al., 2014*). We describe the actin cortical layer as a thin film of a nematic gel under shear and compression by flow (*Salbreux et al., 2009*; *Kruse et al., 2005*; *Prost et al., 2015*). In the x-direction, the time evolution of the local nematic order Q depends on advection (first term on the right hand side in *Equation 1*), compression in gel flow (second term), a process of relaxation to an isotropic configuration possibly via turnover (third term), and a tendency of filaments to align with each other over space (last term, see also *Figure 3a*)

$$\partial_t Q = -v\partial_x Q - \frac{\beta}{2}\partial_x v - \frac{1}{\tau}Q + \frac{\ell^2}{\tau}\partial_x^2 Q. \tag{1}$$

The characteristic time for relaxation to an isotropic state via turnover is determined by $\tau$, while $\beta$ is a dimensionless coefficient that describes how local compression $-\partial_x v$ impacts nematic ordering. Finally, $\ell$ is a length scale that determines the distance over which actin filaments tend to point in the same direction (*Salbreux et al., 2009*). In our experiments, we observe a stationary phase (*Figure 2b*), and the profile of nematic order Q at steady-state is given by

$$Q = -\tau v\partial_x Q - \frac{\beta}{2}\tau\partial_x v + \ell^2\,\partial_x^2 Q. \tag{2}$$

We next determined the theoretical alignment field from the experimental flow field $v$ and compression field $\partial_x v$. We used nonlinear least-square fitting to evaluate parameter values for which the theoretical alignment profile best matched the experimental one. There are five unknowns (three parameters that characterize emergent material properties, $\tau$, $\ell$, and $\beta\tau$, as well as two boundary values of nematic order) but three spatial fields, hence the fitting procedure is significantly constrained and the best fit parameters are uniquely defined in most cases (see Appendix and *Figure 3—figure supplement 1*). For cytokinesis the calculated alignment profile (dashed red lines in *Figure 3b*) best matches the experimentally measured one for $\beta\tau$ =0.64 (0.548, 0.713) min (unless otherwise noted we indicate the median value together with the standard 68% confidence interval of the distribution of the bootstrap data, see Appendix), $\tau$ = 2.3 (1.89, 2.71) min and $\ell$ = 1.7 (1.37, 2.44) µm (*Table 1* and *Table 2*). For pseudocleavage, $\beta\tau$ was similar (0.6 (0.577, 0.7) min), $\tau$ was determined to be smaller than 0.5 min, and $\ell$ = 4.7 (3.12, 6.09) µm larger (*Table 1* and *Table 2*). We note that in both cases, we obtain good agreement between the theoretical and experimental alignment field, allowing us to conclude that compressive gel flow can account for alignment. Further, we observe a small $\tau$ for pseudocleavage and a larger $\tau$ for cytokinesis, which is consistent with the observation that alignment appears concomitant with compression in pseudocleavage but arises with a short delay

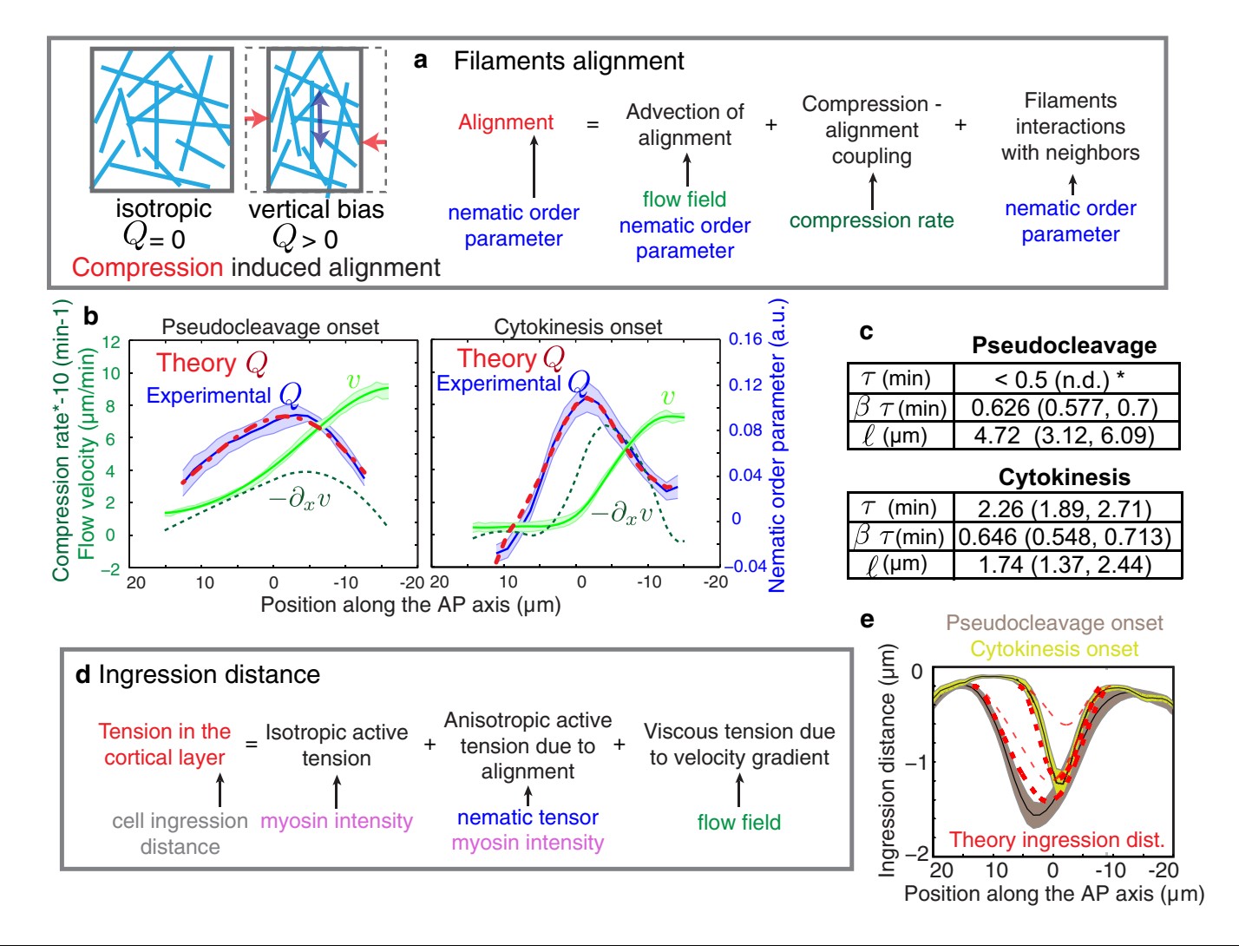

**Figure 3.** Theory predicts an alignment peak. (a) Schematic representation of flow-based alignment. For full theory refer to Appendix. (b) Average AP profiles of gel flow (light green, smoothened), compression rates (dark green) and nematic order parameter (blue) at the time of stationary flow (*Figure 2C*) during pseudocleavage (left, N = 16) and cytokinesis onset (right, N = 32). Error bars represent the standard error of the mean. Dashed red line indicates the respective best-fit theory predictions for the nematic order parameter (blue). (c) Table of the material parameters. See also *Figure 3— figure supplement 1* and *Table 1*, *Table 2*. (d) Illustration of the cortical tension related to the ingression distance. (e) Outer ingression profiles at the onset of pseudocleavage (brown) and cytokinesis (yellow). The theoretical profiles (thick dashed red lines: active tension depends on alignment and can be anisotropic; thin dashed lines: active tension is isotropic) are determined by simultaneous fitting of both the pseudocleavage and cytokinesis datasets with a single fit parameter (see Appendix) using the myosin density, flow velocity and nematic order parameter fields.

The following figure supplements are available for figure 3:

**Figure supplement 1.** Influence of the gel material parameters for the nematic order parameter.

**Figure supplement 2.** Contribution of active alignment to the nematic order parameter profile.

**Figure supplement 3.** Ingression through anisotropic stress generation in the aligned gel.

**Figure supplement 4.** Shape and ingression distance quantification.

**Table 1.** Best fit gel material parameters, bootstrapping. The median values together with the standard 68% confidence bounds of the distribution of the bootstrap data are given.

| | Pseudocleavage | Cytokinesis |
|---|---|---|
| | non RNAi | non RNAi |
| $\tau$, min | < 0.5 (n.d.) [†] | 2.26 (1.89, 2.71) |
| $\beta\,\tau$, min | 0.626 (0.577, 0.7) | 0.646 (0.548, 0.713) |
| $\ell$, μm | 4.72 (3.12, 6.09) | 1.74 (1.37, 2.44) |
| | *nop-1 RNAi* | *nop-1 RNAi* |
| $\tau$, min | n.d. * | < 0.5 (n.d.) [†] |
| $\beta\,\tau$, min | n.d. * | 0.771 (0.694, 2.75) |
| $\ell$, μm | n.d. * | 2.67 (1.7, 11.5) |
| | *ani-1 RNAi* | *ani-1 RNAi* |
| $\tau$, min | < 0.5 (n.d.) [†] | 0.989 (0.555, 1.79) |
| $\beta\,\tau$, min | 1.45 (0.969, 18) [‡] | 0.724 (0.636, 0.929) |
| $\ell$, μm | 12.3 (6.39, 58.4) [‡] | 3.38 (2.97, 6.16) |
| | *mlc-4  (5–7 hr) RNAi* | *mlc-4  (5–7 hr) RNAi* |
| $\tau$, min | < 0.5 (n.d.) [†] | 4.76 (3.67, 6.6) |
| $\beta\,\tau$, min | 0.467 (0.451, 0.503) | 0.637 (0.567, 0.768) |
| $\ell$, μm | 2.75 (2.38, 4.1) | 6.58 (5.31, 6.85) |
| | *mlc-4  (7–9 hr) RNAi* | *mlc-4  (7–9 hr) RNAi* |
| $\tau$, min | n.d. * | < 0.5 (n.d.) [†] |
| $\beta\,\tau$, min | n.d. * | 0.916 (0.82, 1.04) |
| $\ell$, μm | n.d. * | 3.12 (2.49, 4.18) |

*n.d. denotes parameters which could not be determined,

[†]< 0.5 (n.d.) denotes parameters which could not be determined, but could be determined to be below 0.5 (see **Figure 3—figure supplement 1b**),

[‡]denotes values that could be determined but that were not well constrained.

during cytokinesis (**Figure 2d**). We speculate that the changes in physical parameters of the actomyosin cortical layer between pseudocleavage and cytokinesis (higher $\tau$ and smaller $\ell$) reflect the fact that the cell forms a strong and more pronounced ring of aligned filaments during cytokinesis. To conclude, our results are consistent with a scenario in which actin filament alignment arises in a disordered network through compression by flow (**White and Borisy, 1983**; **Salbreux et al., 2009**; **Turlier et al., 2014**).

Until now, we have not considered processes of myosin-based active alignment in our theory (**Vavylonis et al., 2008**). This simplification is probably appropriate for pseudocleavage, but this is less clear for cytokinesis given that there is a clear band of myosin enrichment at the equator when the cell divides (**Figure 1c,d**). We next consider active alignment by myosin, and describe the cortical layer as a thin film of an active nematic gel. For stationary flows the profile of nematic order $Q$ is now given by

$$Q = -\tau v \partial_x Q - \frac{\beta}{2}\tau \partial_x v + \ell^2\, \partial_x^2 Q + \lambda Q, \tag{3}$$

where the right-most term describes active alignment by myosin molecular motors, with $\lambda$ an alignment parameter dependent on the local myosin concentration. In our modified fitting procedure, we now evaluate the respective contributions of myosin-based active and flow-based passive alignment to the stationary Q profile, under the assumption that $\lambda$ is proportional to local myosin concentration. We find that considering active alignment does not increase the overall agreement between calculated and measured alignment profiles for both pseudocleavage and cytokinesis (**Figure 3—**

**Table 2.** Best fit gel material parameters, no bootstrapping. The mean values together with its 95% confidence bounds are given.

| | Pseudocleavage | Cytokinesis |
|---|---|---|
| | **non RNAi** | **non RNAi** |
| $\tau$, min | < 0.5 (n.d.) [†] | 2.34 (1.74, 3.14) |
| $\beta\,\tau$, min | 0.59 (0.513, 0.806) | 0.652 (0.541, 0.764) |
| $\ell$, μm | 5.38 (3.27, 8.85) | 1.69 (1.04, 2.75) |
| | *nop-1 RNAi* | *nop-1 RNAi* |
| $\tau$, min | n.d. * | < 0.5 (n.d.) [†] |
| $\beta\,\tau$, min | n.d. * | 0.738 (0.0.614, 0.862) |
| $\ell$, μm | n.d. * | 1.79 (0.64, 5.03) |
| | *ani-1 RNAi* | *ani-1 RNAi* |
| $\tau$, min | < 0.5 (n.d.) [†] | 1.01 (0.989, 1.03) |
| $\beta\,\tau$, min | 1.31 (0.428, 2.19) | 0.74 (0.588, 0.892) |
| $\ell$, μm | 10.4 (4.91, 21.9) [‡] | 4.08 (2.68, 6.2) |
| | *mlc-4 (5–7 hr) RNAi* | *mlc-4 (5–7 hr) RNAi* |
| $\tau$, min | < 0.5 (n.d.) [†**] | 4.37 (3.08, 6.21) |
| $\beta\,\tau$, min | 0.454 (0.437, 0.47) | 0.614 (0.501, 0.727) |
| $\ell$, μm | 2.24 (1.87, 2.67) | 5.81 (5.02, 6.72) |
| | *mlc-4 (7–9 hr) RNAi* | *mlc-4 (7–9 hr) RNAi* |
| $\tau$, min | n.d. * | < 0.5 (n.d.) [†] |
| $\beta\,\tau$, min | n.d. * | 0.97 (0.931, 1.01) |
| $\ell$, μm | n.d. * | 2.98 (2.88, 3.08) |

*n.d. denotes parameters which could not be determined,

[†]< 0.5(n.d.) denotes parameters which could not be determined, but could be determined to be below 0.5 (see **Figure 3—figure supplement 1b**),

[‡]denotes values that could be determined but that were not well constrained. Caption for **Videos 1Videos 1 to 7**

figure supplement 2). Notably, the contribution of active alignment to pseudocleavage is insignificant, while active alignment contributes to enhancing compression-induced alignment during cytokinesis (*Figure 3—figure supplement 2b*; see Appendix), albeit to a small degree. We conclude that compression by flow and not myosin-based active alignment is the driving force of ring formation in both pseudocleavage and cytokinesis.

A key prediction from our model is that reducing flow speeds and compression rates should give rise to less filament alignment and a possible inhibition of pseudocleavage furrow formation. To test this prediction, we performed weak-perturbation RNAi experiments (*Naganathan et al., 2014*) of

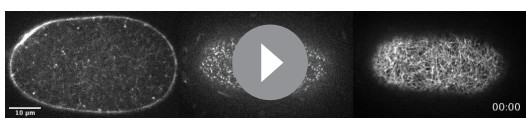

**Video 5.** Actomyosin gel dynamics in the *C. elegans* zygote after 24 hr *nop-1(RNAi)*. Cortical and medial planes of an embryo expressing both Lifeact::mKate2 and endogenous NMY-2::GFP. Left panel, medial plane Lifeact:mKate2, center, cortical NMY-2::GFP, right panel, cortical Lifeact:mKate2 (min:s).

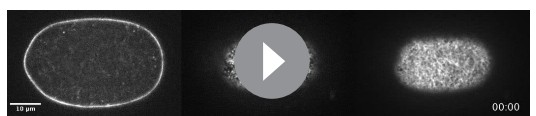

**Video 6.** Actomyosin gel dynamics in the *C. elegans* zygote after 26 hr *ani-1(RNAi)*. Cortical and medial planes of an embryo expressing both Lifeact::mKate2 and endogenous NMY-2::GFP. Left panel, medial plane Lifeact:mKate2, center, cortical NMY-2::GFP, right panel, cortical Lifeact:mKate2 (min:s).

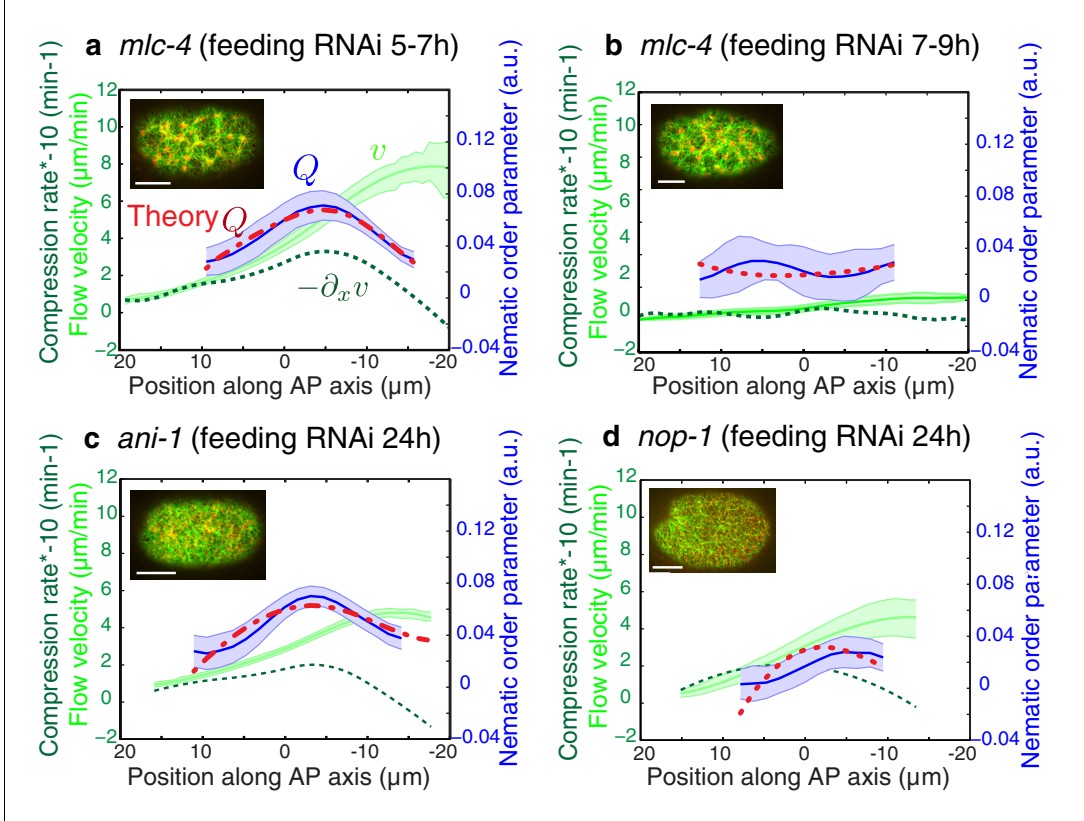

**Figure 4.** Flows and compression are required to generate alignment. (**a-d**) Average AP profiles of gel flow (light green, smoothened), compression rates (dark green) and nematic order parameter (blue) at the time of stationary flow during pseudocleavage onset for *mlc-4* 5–7 hr and 7–9 hr, *nop-1* and *ani-1 RNAi*. Error bars represent the standard error of the mean (N = 17, 14, 10, 22 for a-d, respectively). Dashed red line indicate the respective best-fit theory predictions for the nematic order parameter (blue). For *ani-1* and *mlc-4* (5–7 hr), $\tau$ is small and is determined to be smaller than 0.5 min. For *nop-1* and *mlc-4* (7–9 hr), theory profiles were generated using the parameters obtained for the unperturbed non-RNAi pseudocleavage, since insufficient compression rates did not constrain the physical parameters. Scale bar, 10 µm.

The following figure supplements are available for figure 4:

**Figure supplement 1.** Cytoskeletal organization under RNAi perturbation.

**Figure supplement 2.** Flows, compression and alignment at cytokinesis onset.

the regulatory myosin light chain of non-muscle myosin (*mlc-4(RNAi)*) (*Craig et al., 1983*) to mildly reduce actomyosin flow speeds without significant changes in overall actomyosin organization (*Figure 4—figure supplement 1*). Short feeding times (5–7 hr) lead to a small reduction in gel flow and compression rates (*Figure 4a*), but actin filaments still aligned in the central region and the pseudocleavage furrow still formed and ingressed. Consistent with our hypothesis, longer feeding times (7–9 hr) lead to a complete abolishment of cortical flow and a loss of both actin alignment and pseudocleavage furrow (*Figure 4b*). Actomyosin foci are not required for compression-based alignment since anillin- depleted embryos (*ani-1 (RNAi)*, 24 hr) still show significant alignment under compression in the absence of these dense foci (*Figure 4c*) (*Maddox et al., 2005*; *Piekny and Maddox, 2010*; *Tse et al., 2011*).

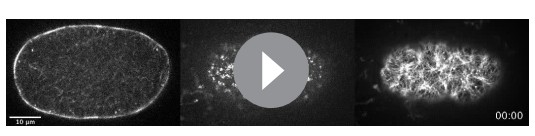

**Video 7.** Actomyosin gel dynamics in the *C. elegans* zygote after 14 hr *mlc-4(RNAi)*. Cortical and medial planes of an embryo expressing both Lifeact::mKate2 and endogenous NMY-2::GFP. Left panel, medial plane Lifeact:mKate2, center, cortical NMY-2::GFP, right panel, cortical Lifeact:mKate2 (min:s).

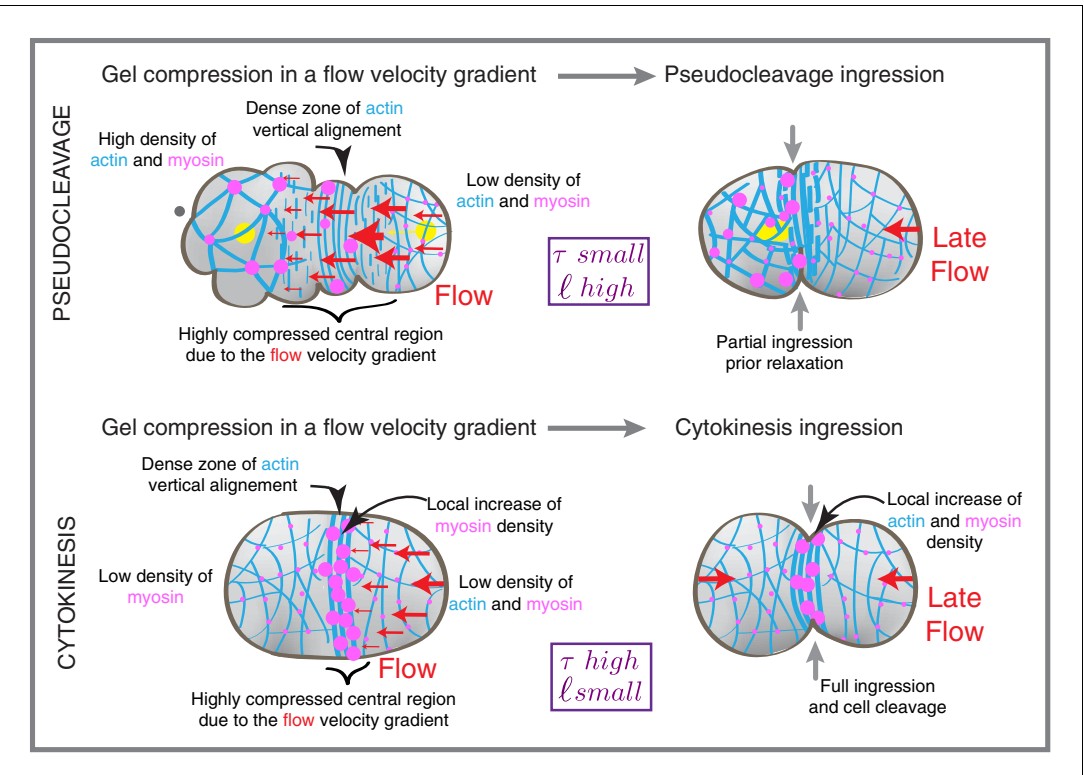

**Figure 5.** Compression drives alignment for furrowing. Schematic illustration of flows, cytoskeletal organization and cell shape changes during pseudocleavage and at the onset of cytokinesis.

Reduced flow speeds and compression rates can also account for the absence of alignment and pseudocleavage in *nop-1(RNAi)* embryos, in which RhoA-dependent processes are affected and pseudocleavage formation abrogated (*Figure 4d*, *Figure 4—figure supplement 1* and *2*) (*Tse et al., 2012*; *Rose et al., 1995*; *Zhang and Glotzer, 2015*). Overall, these experiments lead us to conclude that reducing gel flow and compression rates affects alignment in a manner that is consistent with *Equation 2*. Finally, a complete loss of flow and compression abolishes pseudocleavage entirely (*Figure 4b*;d and *Figure 4—figure supplement 1*). Taken together, our results suggest that the pseudocleavage furrow arises as a by-product of compressive flows, mechanically aligning actin filaments into a ring even in the absence of localized RhoA activation.

In this context we wondered if the cortex is specifically modified to favor ring formation in cytokinesis. We used our fitting procedure to determine the parameters that characterize emergent material properties of the cortex under *mlc-4(RNAi)*. Notably, we observe that 7–9 hr of *mlc-4(RNAi)* leads to a reduction of the alignment relaxation time scale τ during cytokinesis. The value obtained is similar to the alignment relaxation time scale during pseudocleavage under non-RNAi conditions (*Table 1* and *Table 2*), leading us to speculate that an increase in myosin activity via Rho signaling is responsible for the observed increase of the characteristic relaxation time τ in cytokinesis. Importantly, increasing τ allows the cell to form a pronounced ring of aligned filaments by compressive cortical flow during cytokinesis, as predicted by theory (*Figure 3—figure supplement 1*).

Finally, we sought to find support from theory that aligned filaments in the equatorial region drive anisotropic stress generation (*Mayer et al., 2010*) to form an ingression (*White and Borisy, 1983*; *Salbreux et al., 2009*; *Turlier et al., 2014*). By use of a theory that connects anisotropic active stress generation in the active nematic gel to changes in cell shape (see Appendix and *Figure 3d*) we show that we cannot appropriately account for the observed ingression profiles unless we consider anisotropic active stress generation in an aligned gel (*Figure 3e*, *Figure 3—figure supplement 3*). This suggests that anisotropic active stress generation in a network of aligned actin filaments is important for forming an ingression.

To conclude, we find that compression by flow drives actomyosin ring formation in both pseudo-cleavage and cytokinesis, as originally proposed by White and Borisy (*White and Borisy, 1983*) (*Figure 5*). For this, we provide a general framework for determining emergent material parameters of the actomyosin gel. This characterizes flow-alignment coupling and allows for capturing essential aspects of the mechanics of furrow generation. Our analysis of the pseudocleavage ingression in *C. elegans* demonstrates that constricting rings can form in the absence of equatorial RhoA activation. This reveals that cortical flow functions as a central organizer of network architecture, mechanically aligning filaments to form a constricting furrow in the equatorial region without a localized increase of actin nucleation (*Figure 2—figure supplement 5,f*) and myosin contractility (*Figure 1b-d*). Finally, it will be interesting to investigate if a cortex that is aligned by compressive flow favors the recruitment of specific actin binding proteins such as bundling or motor proteins (*Robinson et al., 2002*), comprising an interesting mechanism of positive feedback for stabilizing and enhancing the ring during cytokinesis. Our work highlights that determining emergent material properties of actomyosin is important for understanding cytokinesis, and a challenge for the future is to link emergent material properties at the larger scale with molecular mechanisms (*Mendes Pinto et al., 2013*; *Eggert et al., 2006*).

## Materials and methods

### *C. elegans* strains

Existing reagents did not allow for detailed and long-term imaging of actin filaments in embryos, which is critical for reliably quantifying their orientation. Hence, we generated a new Lifeact transgenetic line with enhanced actin filament labeling (SWG001). A codon optimized for *C. elegans* far-red fluorophore, mKate2 kindly shared by Henrik Bringman (*Redemann et al., 2011*) (26 kDa, 588/633 nm) was added to the actin probe with a linker (66 bp) and cloned into MOSCI vector containing unc119 rescue gene (Hyman Lab) under the control of the mex-5 promoter. Stable integration was obtained by bombardment of this plasmid in DP38 strain. This strain was then backcrossed with N2. In this line, we observed within the cortical plane bright actin filament labeling with a high signal-to-noise ratio and little photobleaching. Importantly, overall cortical organization and flow dynamics appeared to not be affected by this reagent, foci lifetime, spacing, cortical flow velocities were similar to previous measurement with other fluorescent lines. A dual colored transgenic line was obtained in order to image simultaneously actin and myosin dynamics by crossing Lifeact::mKate2 strain with LP133 (NMY-2::GFP obtained by CRISPR (*Dickinson et al., 2013*), SWG007). The RhoA biosensor used for *Figure 1* was developed by the Glotzer lab (GFP::AHPH, *C. elegans* strain MG617, *Tse et al., 2012*). This sensor consists of GFP fused to the C-terminal portion of *C. elegans* anillin, which contain its conserved region (AH) and pleckstrin homology (PH) domain. It lacks the N-terminal myosin and actin-binding domains but retains its RhoA-binding domain. The strains expressing CYK-1:GFP (SWG004) and PLST-1:GFP (SWG005) were obtained by the insertion of a GFP tag at the endogenous locus in their C-terminal region using CRISPR according to Dickinson et al (*Dickinson et al., 2013*). *C. elegans* worms were cultured on OP50-seeeded NGM agar plates as described (*Brenner, 1974*).

### Microscopy

L4 mothers were maintained overnight at 20°C before dissection in M9 buffer. For imaging, one-cell embryos were mounted on 2% agar pads, thereby slightly compressed to increase the cortical surface visible in a confocal plane. Images were acquired with spinning disk confocal microscope (Zeiss C-Apochromat, 63X/1.2 NA or 100X/1.42 NA, Yokogawa CSU-X1 scan head and Hamamatsu Orca-Flash4.0 camera) every 2 s on two or three different planes (one or two at the bottom for a cortical section, and one 12 μm above for a medial section of the embryo) . For 3D acquisitions, Z-stacks were acquired every 0.2 μm. We assume the shape to be rotationally symmetrical to the longer axis, thus the outline of one medial section of the embryo reflects faithfully cell shape changes and ingression distance.

## Gene silencing by RNA interference

RNAi experiments were performed by feeding (*Timmons et al., 2001*). L4 worms were grown at 20℃ on feeding plate (NGM agar containing 1 mM isopropyl-β-D-thiogalactoside and 50 μg ml$^{-1}$ ampicillin) for the required number of hours before imaging.

## Correlation

Spatiotemporal correlation functions were calculated for all time points during the stationary period, in the 30 μm central region of the embryo. The plot of the spatial correlation only (1D correlation function) is shown in *Figure 2c* between the compression, alignment and ingression data sets. Peak value positions are obtained by a Gaussian fitting procedure and allow the calculation of positional differences between the different data sets. Because the flow field appears stationary, we expect this distance between peaks of compression, alignment and ingression to arise due to temporal delays between these events in the reference frame co-moving with the cortical flows. We thus translated the characteristic distances between stationary peaks to temporal delays using the average flow velocity in the $-10$ μm to 0 central region (6.16 ± 1.23 μm/min for pseudocleavage and 4.01 ± 1.92 μm/min for cytokinesis) (*Figure 2d*).

## Acknowledgements

We thank Pierre Gönczy, Caren Norden, Enrique De La Cruz, Michel Labouesse, Tony Hyman and Marino Zerial for insightful comments on the manuscript. We thank Jean-Francois Joanny and Jacques Prost for invaluable discussions of some of the ideas presented here. We thank Ralph Böhme for his work on image quantification validation. A-C Reymann is supported by the Human Frontier Science Program LT000926/ 2012. AE is supported by the excellence initiative of the German Research Foundation (DFG-GSC 97/3). SWG was supported by the DFG (SPP 1782, GSC 97, GR 3271/2, GR 3271/3, GR 3271/4), the European Research Council (grant No 281903), ITN grants 281903 and 641639 from the EU, and the Human Frontier Science Program (RGP0023/2014). GS acknowledges support by the Francis Crick Institute which receives its core funding from Cancer Research UK (FC001317), the UK Medical Research Council (FC001317), and the Wellcome Trust (FC001317). The Max Planck Society also supported this work.

## Additional information

### Funding

| Funder | Grant reference number | Author |
| --- | --- | --- |
| Human Frontier Science Program | LT000926/ 2012 | Anne-Cecile Reymann |
| Deutsche Forschungsgemeinschaft | DFG-GSC 97/3 | Anna Erzberger |
| Max-Planck-Gesellschaft | | Guillaume Salbreux Stephan W Grill |
| Cancer Research UK | FC001317 | Guillaume Salbreux |
| Medical Research Council | FC001317 | Guillaume Salbreux |
| Wellcome Trust | FC001317 | Guillaume Salbreux |
| Human Frontier Science Program | RGP0023/2014 | Stephan W Grill |
| European Research Council | 281903 | Stephan W Grill |
| Deutsche Forschungsgemeinschaft | SPP 1782 | Stephan W Grill |
| Deutsche Forschungsgemeinschaft | GR 3271/2 | Stephan W Grill |
| Deutsche Forschungsgemeinschaft | GSC 97 | Stephan W Grill |

| Deutsche Forschungsgemeinschaft | GR 3271/3 | Stephan W Grill |
| Deutsche Forschungsgemeinschaft | GR 3271/4 | Stephan W Grill |
| European Research Council | 641639 | Stephan W Grill |

The funders had no role in study design, data collection and interpretation, or the decision to submit the work for publication.

## Author contributions

A-CR, Performed the experiments, The presented ideas, analysis and the theory were developed together by all authors, Conception and design, Drafting or revising the article, Contributed unpublished essential data or reagents; FS, The presented ideas, analysis and the theory were developed together by all authors, Developed the mathematics and fitting procedures for the alignment theory, Drafting or revising the article; AE, The presented ideas, analysis and the theory were developed together by all authors, Developed the mathematics and fitting procedures for the ingression theory, Drafting or revising the article; GS, SWG, The presented ideas, analysis and the theory were developed together by all authors, Conception and design, Drafting or revising the article

## Author ORCIDs

Anne-Cecile Reymann, [iD] http://orcid.org/0000-0002-0517-5083

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

## Appendix

### Image analysis and quantification

Image segmentation was done on the medial plane acquisition using the JFilaments plugin of ImageJ (*Smith, 2010*). Ingression levels along the AP axis were calculated using the position of the eggshell as a reference. The thickness of the extra-embryonic layer (*Johnston and Dennis, 2012*) (0.2 μm) was subtracted prior to comparison with the theoretical predictions. We considered here only the outer ingression without the membrane overlap zone (as shown in *Figure 2b* and *Figure 3—figure supplements 3* and *4*). The outer ingression level is a good indicator of cell shape changes at the onset of the formation of an ingression furrow (before deep membrane overlap region is actually extended), the period of interest for this study. Background fluorescence was subtracted for fluorescence intensity quantification, and normalized to maximum intensity for the RhoA biosensor.

Flow velocities were obtained by Particle Image Velocimetry tracking on Lifeact::mKate2 timelapse acquisitions (*Thielicke and Stamhuis, 2014*) (PIV template size is 0.84 μm (8px)). The x-axis is defined as the posterior-anterior axis and is oriented towards the anterior. Filament orientation was quantified for images in *Figure 1—figure supplement 1* using the SteerableJ plugin for Fiji. To quantify the nematic order parameter characterizing filament orientation, we used an indirect method based on Fourier Transform analysis of fluorescence image intensity. This analysis relies on the fact that the magnitude of the Fourier Transformed image encodes the geometric characteristics of the spatial domain image, therefore reflecting its main direction or orientation pattern. Note that we do not quantify unique filament orientation but an average orientation at the micrometer scale due to high density of filaments and the presence of bundles. Quantification was done as followed. The image was segmented in square templates of 3.15 μm (N = 30 px). Denoting the image fluorescence intensity $I(x, y)$ at position $(x, y)$ in pixels, the Fourier Transform $F_T$ of the intensity was then calculated according to

$$F_T(q_x, q_y) = \sum_{x,y} I(x,y) e^{-i \frac{2\pi(q_x x + q_y y)}{N}}.$$ (A1)

The normalized squared modulus of the Fourier transform is obtained by

$$G(q_x, q_y) = |F_T(q_x, q_y)|^2$$ (A2)

Using then polar coordinates in Fourier space $q, \theta$ with $q_x = q \cos \theta$ and $q_y = q \sin \theta$, a normalized angular function $h$ is obtained as

$$H(\theta) = \sum_{q=q_{min}}^{q=q_{max}} G(q, \theta), \quad h(\theta) = \frac{H(\theta)}{\sum_{\theta'} H(\theta')},$$ (A3)

with $q_{min} = 3$ and $q_{max} = 12$. The components of the nematic tensor

$$\mathbf{Q} = \begin{pmatrix} Q_{xx} & Q_{xy} \\ Q_{yx} & Q_{yy} \end{pmatrix} = \begin{pmatrix} -Q & Q_{xy} \\ Q_{xy} & Q \end{pmatrix}$$ (A4)

were then evaluated according to

$$Q_{xx} = -\sum_{\theta} h(\theta) \left( (\cos\theta)^2 - \frac{1}{2} \right) \tag{A5}$$

$$Q_{xy} = -\sum_{\theta} h(\theta) \cos\theta \sin\theta. \tag{A6}$$

where the negative sign is introduced to take into account the rotation in order introduced by the Fourier transform. $Q_{xx}$>0 or Q<0 indicates positive alignment parallel to the x axis, while Q>0 indicates alignment parallel to the y axis. $Q_{xy} = Q_{yx}$>0 indicates alignment along the diagonal $y = x$, and $Q_{xy}$<0 indicates alignment along the diagonal $y = -x$.

Embryos were aligned in time (see below) and in space according to the embryo center. For averaging purposes only the embryos for which the direction of the polarizing flow corresponded to the long axis of the embryos were taken into account, and embryos that polarized from the side were discarded. For alignment with respect to time, temporal shifts were calculated by fitting an error function to the anterior-posterior velocity $v_x$ evolution in time obtained at a fixed position in the posterior region (an example is shown in **Figure 2— figure supplement 1b**). Flow velocities and nematic order parameters were then averaged in space and time, and over an ensemble of embryos. We identified the time period of approximate stationarity from these average graphs (see also **Figure 2b** and **Figure 2—figure supplement 1**). As a control, choosing a period of a minute just prior to onset of furrow ingression for each embryo individually, and then determining the ensemble average of these graphs gave very similar results (data not shown).

## Validation of nematic order parameter quantification on artificial images

To check that our quantification method of the nematic order parameter is meaningful and reliable, we performed tests with synthetic images of filamentous networks generated with Matlab. These artificial images allowed us to modulate several parameters: the filament density, their intensity as well as the signal to noise ratio. We proceed as follows for the generation of these images: each filament has a random length taken from a normal distribution with mean and standard deviation 50±20 μm, a random starting point is selected in the image, from which the filament elongates. During elongation, a random small variation in the direction of growth is introduced, as well as in the local filament brightness. Variations in order are introduced by selecting the initial direction of growth of each filament $\alpha$. The angle $\alpha$ takes values between 0 and $\pi$ and is selected from a probability $P(\alpha)$ which can either be uniform ($P_{unif}(\alpha) = 1/\pi$) or built such as to favor the vertical direction of initial growth around the orientation $\alpha = 0$. The anisotropy in the angle distribution is modulated by a parameter B: for B=0 the network is isotropic, and becomes more anisotropic for increasing value of B. The final synthetic images of the actin network are then blurred by a Gaussian filtering before some background noise is added to reflect the impact of the point-spread function of the microscope (**Figure 2—figure supplement 4a**). **Figure 2—figure supplements 3** and **4** shows the result of quantification of the nematic order parameter using such synthetic images. Our measured nematic order parameter reflects the anisotropy in the artificial networks (**Figure 2—figure supplement 3d**), demonstrates that increasing filaments density does not significantly impact the quantification, and reveals that a good signal to noise ratio is required. Note however that our measured order parameter can only be viewed as an approximation of the actual nematic order parameter of the network, but does capture general features of the state of alignment.

## Changing flow direction changes alignment direction

To rule out the possibility that the ovoid zygote shape or an unknown pre-established positional cue determines the location and the direction of filament alignment, we analyzed embryos in which flow was initiated at the side and far from the posterior pole of the embryo. Such a situation arises in rare cases due to off-site sperm entry (*Goldstein et al., 1993*). Here, cortical flow is initially not aligned with the long axis of the egg (*Hird and White, 1993*; *Rappleye et al., 2002*). If filament alignment indeed arises through compression by flow, we would expect filaments to still align in a direction that is perpendicular to the axis of compression and the direction of flow even when flows are not aligned with the AP axis. Indeed, we observed initial actin filaments alignment along the long axis of the egg and thus in a direction determined by compression in cases where flow was initiated at the side of the embryo (*Figure 1—figure supplement 1d* and *Video 4*). We conclude that actin filament alignment follows cortical flow and compression and is independent of the elongated shape of the embryo.

## Active nematic gel theory

The nematic order parameter for a two-dimensional liquid crystal is defined in a Cartesian frame as (*De Gennes and Prost, 1993*) $Q_{ij} = \langle n_i n_j - \frac{1}{2}\delta_{ij}\rangle$ and can be used to describe actin filament alignment. The brackets indicate the local averages over all filament bundles at a given position, and $n_i$, $n_j$ are the components of the unit vectors along which each individual filament bundle is aligned. Note that we do not quantify unique filament orientation but an average orientation at the micrometer scale due to high density of filaments and the presence of bundles. The nematic tensor is traceless, such that $Q_{yy} = -Q_{xx} = Q$.

We discuss here a theoretical description for the dynamics of the nematic tensor $Q_{ij}$. We neglect here the curvature of the embryo and write equations in an effective Cartesian frame. We use Einstein's notation for summation, such that repeated indices are summed. By construction the nematic tensor is traceless and symmetric (*Equation A4*). The constitutive equation in two dimensions for the dynamic of the nematic order parameter in active fluid reads:

$$\frac{D}{Dt}Q_{ij} = -\frac{1}{\gamma}\frac{\delta F}{\delta Q_{ij}} + \lambda_0(c)\,\Delta\mu Q_{ij} + \beta(v_{ij} - \frac{1}{2}v_{kk}\delta_{ij}), \tag{A7}$$

where $\frac{D}{Dt}$ denotes the comoving and corotational derivative, and where the symmetric part of the velocity gradient tensor is defined as

$$v_{ij} = \frac{1}{2}(\partial_i v_j + \partial_j v_i), \tag{A8}$$

with $\partial_i$ the spatial derivative on the surface, and we assume that the surface is not deforming. The comoving and corotational derivative of the nematic tensor can be written as

$$\frac{D}{Dt}Q_{ij} = \frac{\partial Q_{ij}}{\partial t} + v_k(\nabla_k Q_{ij}) + [w_{ik}Q_{kj} + w_{jk}Q_{ki}] \tag{A9}$$

Note that this equation includes the effect of advection, since it couples the velocity field to the spatial derivatives of the nematic order parameter. The rotation of the nematic order parameter by flow is captured by the coupling of the nematic order parameter to the anti-symmetric part of the velocity gradient tensor, given by

$$w_{ij} = \frac{1}{2}\left(\partial_i v_j - \partial_j v_i\right).$$ (A10)

In **Equation (A7)**, the last term describes a flow-coupling alignment generated by velocity gradients, with $\beta$ a dimensionless factor. The second term proportional to $\Delta\mu$, the chemical potential of ATP hydrolysis, corresponds to active alignment effects introduced by motors with concentration c in the network. For $\lambda_0 > 0$, myosin molecular motors tend to enhance alignment of filaments, while for $\lambda_0 < 0$, myosin motors tend to decrease filament ordering. We will discuss separately the case where $\lambda_0 = 0$ (no myosin-based active alignment) and where $\lambda_0 \neq 0$ (with myosin-based active alignment). The first term proportional to $\frac{\delta F}{\delta Q_{ij}}$ depends on the free energy $F$ of the network of filament and captures the relaxation of the system to thermodynamic equilibrium. We assume the following form for the free-energy of the system

$$F = K \int \left[\frac{1}{2}Q_{ij}Q_{ij} + \frac{\ell^2}{2}\nabla_k Q_{ij}\nabla_k Q_{ij}\right] dS,$$ (A11)

where K is a inverse nematic susceptibility, characterizing the tendency of the network to return to isotropic orientation of filaments. Here, $\ell$ is a characteristic length scale below which filaments are coherently aligned. This choice of free energy with K > 0 implies that filaments do not spontaneously align, and that alignment is generated only by velocity gradients in **Equation (A7)**. The derivative of the free energy with respect to the nematic tensor reads:

$$\frac{\delta F}{\delta Q_{ij}} = K\left(Q_{ij} - \ell^2 \Delta Q_{ij}\right),$$ (A12)

with $\Delta$ the Laplacian operator, such that **Equation (A7)** can be rewritten

$$\frac{D}{Dt}Q_{ij} = -\frac{1}{\tau}\left(Q_{ij} - \ell^2 \Delta Q_{ij}\right) + \beta\left(v_{ij} - \frac{1}{2}v_{kk}\delta_{ij}\right) + \lambda_0(c)\Delta\mu\, Q_{ij},$$ (A13)

with $\tau = \gamma/K$ the characteristic time of the dynamics of the nematic order parameter's relaxation.

We consider here the cortex to be rotationally symmetric around the AP axis. As a consequence we can neglect derivatives along the direction perpendicular to the AP axis direction, here denoted y. We also neglect effects introduced by the cortex velocity around its axis symmetry, which is smaller than the velocity along the AP axis (see **Figure 2—figure supplement 1**). We then obtain the following simplified equation for the components of the nematic tensor, varying along the AP direction x:

$$\partial_t Q_{xx} = -v_x \partial_x Q_{xx} + \frac{\beta}{2}\partial_x v_x - \frac{1}{\tau}Q_{xx} + \frac{\ell^2}{\tau}\partial_x^2 Q_{xx} + \lambda_0(c)\Delta\mu Q_{xx},$$ (A14)

$$\partial_t Q_{xy} = -v_x \partial_x Q_{xy} - \frac{1}{\tau}Q_{xy} + \frac{\ell^2}{\tau}\partial_x^2 Q_{xy} + \lambda_0(c)\Delta\mu\, Q_{xy}.$$ (A15)

At steady state, **Equation (A14)** reduces to **Equation (2)** in the main text for the case that $\lambda_0 = 0$, and **Equation (3)** for $\lambda_0 \neq 0$, with $Q = Q_{yy} = -Q_{xx}$ and $\lambda = \tau\lambda_0(c)\Delta\mu$. For $\lambda > 1$, the isotropic state becomes unstable and filaments spontaneously align; we assume in the following $\lambda < 1$. Note that **Equation (A15)** yields $Q_{xy} = 0$. The first term on the right hand side of **Equations (2) and (3)** describes advection of alignment by flow, the second term accounts for the coupling between compression and alignment, the third term captures that actin filaments tend to be aligned with their neighbors (as represented in **Figure 3a**). The fourth

term on the right hand side of **Equation (3)** captures active alignment driven by myosin activity.

Note that considering active alignment by myosin motor proteins results in the addition of an extra term where nematic order is proportional to myosin. This implies that active alignment only enhances or decreases the degree of ordering, but on its own cannot set the main orientation of filaments. Hence the orientation of alignment is still dictated by compressive flows. Using this additional term for an active nematic fluid allows us to estimate the contribution of active alignment by myosin to the experimental nematic order parameter profile, which is insignificant for pseudocleavage and is low for cytokinesis (**Figure 3—figure supplement 2**).

Note also that the product $\beta\tau$ determines the coupling between compression and alignment in stationary flow, and controls the height of the nematic order parameter peak (**Figure 3—figure supplement 1**). The length scale $\ell$ influences the width of the alignment peak, while the time scale of relaxation $\tau$ controls the distance between the compression peak and the alignment peak (**Figure 3—figure supplement 1**).

## Fitting procedure for nematic order parameter profiles

For the fits shown in **Figures 3b**, **4** and **Figure 3—figure supplements 1** and **2**, we use experimental data for $Q_{yy}(x)$ and $v_x(x)$ measured along the AP $x$-axis and averaged as described previously during the stationary flow period and over several embryos. In order to verify whether **Equations (2) and (3)** in the main text can account for experimentally measured profiles of the nematic order parameter, we proceeded as follows.

Continuous velocity profiles along the AP axis $v_x(x)$ are obtained by fitting spline functions to measurements of cortical velocity. The spline functions are chosen to be piecewise third order polynomial functions, with imposed continuity of the function and its first derivative. The continuous velocity profiles are used to obtain the corresponding velocity gradient $\partial_x v_x(x)$ (**Figure 3b**). **Equations (2) and (3)** can then be numerically integrated using this velocity gradient. To integrate them, we left the values of $Q_{yy}(x)$ and $\partial_x Q_{yy}(x)$ at the left boundary as free fitting parameters $C_1$ and $C_2$. In addition, we specified the parameters $\beta\tau$, $\tau$ and $1/\ell^2$, as well as $\lambda'\tau$ ($\lambda = \lambda'\tau \, c(x)/c_0$, with $c$ the myosin concentration and $c_0$ a reference concentration taken to be the maximum measured concentration) when myosin-based active alignment is taken into account. For each theoretical curve reported in the paper, the parameters $\beta\tau$, $\tau$, $1/\ell^2$ and $\lambda'\tau$ as well as the boundary values of $Q_{yy}$ and $\partial_x Q_{yy}$ are adjusted with a least squares fitting procedure to match the theoretical profile of $Q_{yy}(x)$ to the experimental profile. Uncertainties for the fitting parameters are obtained as the square root of the diagonal terms of the covariance matrix for the parameter estimates.

Negative values for $\tau$ or $\ell^2$ are not allowed in our description as we assume that the cortex is in the isotropic phase. We impose positivity of these parameters by introducing renormalized parameters $\tau = e^a$ and $\ell^2 = e^b$, fitting the new parameters $a$ and $b$ and transforming them back to obtain the physical parameters.

To check for consistency of fitting results and to provide an alternative means of estimating confidence intervals, we implemented a bootstrap method. Once the fitting procedure is performed, we generate 100 new sets of data points by reshuffling randomly the residuals of the fit and adding them to the profile of nematic order $Q_{yy}(x)$ obtained from the original fit, evaluated at the positions of the data points. The new data sets generated by this method are used to perform new fits and obtain a distribution of fitting parameters. Importantly, we found the original fitting parameters to be placed within the 68% confidence interval of the distribution of the new data set (compare **Table 1** and **Table 2**).

For pseudocleavage, $\tau$ was found to be small and the data did not allow for its precise quantification, but fixing its value to times below 0.5 min allows for essentially equally good

matches between theory and experiment without significant changes to the other parameters (*Figure 3—figure supplement 1*). In addition, note that increasing the time τ tends to increase the degree of alignment for a given flow and compression profile (*Figure 3—figure supplement 1*). Furthermore, decreasing the characteristic length $\ell$ gives rise to a 'sharper' and more confined alignment peak (*Figure 3—figure supplement 1*).

## Tension induced cell shape changes

We propose here a physical theory describing how the tension in the cortical layer governs the shape of the cell during furrow initiation. We describe the cortical layer as an axisymmetric, two-dimensional surface. In the following we use the conventional notation in differential geometry, using the indices i, j to refer to the chosen coordinates parameterizing the surface, with upper and lower indices representing contravariant and covariant coordinates respectively. Internal tensions in the cortical surface are described by a tension tensor $t^i_j$, which has passive and active contributions. We assume that the actin network behaves as a viscous gel at the timescale relevant for furrow initiation, and we characterize it by bulk and shear viscosities $\eta_b$ and $\eta_s$. In addition, the chemically driven activity of myosin motors embedded in the actin network generates contractile stresses in the cortical layer. In disordered actin networks, myosin-dependent active stresses are isotropic. The alignment of actin filaments however may introduce a preferred direction for motor-filament interactions, giving rise to anisotropies in the cortical stress (*Salbreux et al., 2009*). The constitutive relation for the tension tensor in the cortex then reads

$$t^i_j = \eta_s\left(v^i_j - \frac{1}{2}v^k_k\delta^i_j\right) + \eta_b v^k_k\delta^i_j + \zeta\delta^i_j + t^i_{j,\text{nematic}} \, . \tag{A16}$$

where $v^i_j$ is the symmetric velocity gradient tensor on the surface (*Equation A26*). Here, $\zeta$ represents the isotropic part of the active tension generated by myosin molecules, and $t^i_{j,\text{nematic}}$ is the part of the active tension which depends on the nematic order parameter $Q^i_j$.

A confining rigid eggshell surrounds the *C. elegans* zygote (see *Figure 3—figure supplement 3* and *4*). When the embryo forms an ingression during pseudocleavage or cytokinesis, there are two contact lines between the cortex and the eggshell at the two ends of the ingression zone, which in a midplane cut through the embryo are represented by to contact points $s_1$ and $s_2$. While outside of the ingression zone, the cell is constrained by the eggshell, the shape taken between the contact points is determined by actomyosin-intrinsic forces and the pressures in the surrounding fluids (see *Figure 3—figure supplement 3d*). Considering the balance of forces in the direction normal to the cell surface in this region thus allows calculating the level of ingression. The forces arising from the viscous and active tension in the curved cortical surface must balance the fluid pressure acting from the inside $P_{\text{int}}$ and the outside $P_{\text{ext}}$ of the embryo, giving rise to the Young-Laplace equation

$$C_{ij}t^{ij} = P_{\text{int}} - P_{\text{ext}} \, , \tag{A17}$$

where $C_{ij}$ is the curvature tensor. We use here *Equation (A17)* to calculate the shape of the embryo. For completeness, the tangential force balance reads, assuming that an external friction force is acting on the cortex,

$$\nabla_i t^{ij} = \bar{\gamma}v^j, \tag{A18}$$

with $\bar{\gamma}$ a friction coefficient and $\nabla_i$ the covariant derivative on the surface. In principle *Equation (A18)* can be solved for the velocity flowfield; for simplicity we do not attempt to solve this equation here and instead use the experimentally measured flowfield. We

parameterize the axisymmetric shape by the arclength and angle coordinates s and $\phi$ (see *Figure 3—figure supplement 3c*), such that a point on the ingressed part of the embryo is given by

$$\mathbf{X}(s,\phi) = -x(s)\mathbf{e}_x + r(s)\sin\phi\,\mathbf{e}_y + r(s)\cos\phi\,\mathbf{e}_z \qquad (A19)$$

in the cartesian basis $\mathbf{e}_x,\ \mathbf{e}_y, \mathbf{e}_z$. The tangent vectors at each point on the surface are given by

$$\mathbf{e}_s = \partial_s\mathbf{X} = \begin{pmatrix} -\partial_s x \\ \sin\phi\,\partial_s r \\ \cos\phi\,\partial_s r \end{pmatrix}, \qquad (A20)$$

and

$$\mathbf{e}_\phi = \partial_\phi\mathbf{X} = \begin{pmatrix} 0 \\ r\cos\phi \\ -r\sin\phi \end{pmatrix}. \qquad (A21)$$

The vector normal to the surface is given by

$$\mathbf{n} = \frac{\mathbf{e}_s\,\times\,\mathbf{e}_\phi}{(|\mathbf{e}_s\,\times\,\mathbf{e}_\phi|)} = \begin{pmatrix} -\partial_s r \\ -\sin\phi\,\partial_s x \\ -\cos\phi\,\partial_s x \end{pmatrix}. \qquad (A22)$$

We impose that s is an arclength parameter, such that $|\mathbf{e}_s| = \partial_s r^2 + \partial_s x^2 = 1$. Defining $\psi$ as the angle between $\mathbf{e}_s$ and the plane normal to the x-axis, we have the relations, $\partial_s r = \cos\psi$ and $\partial_s x = -\sin\psi$. The metric tensor of the surface is given by

$$g_{ij} = \mathbf{e}_i\cdot\mathbf{e}_j = \begin{pmatrix} 1 & 0 \\ 0 & r^2 \end{pmatrix}, \qquad (A23)$$

and the curvature tensor is defined as

$$C_{ij} = -\mathbf{n}\cdot\partial_i\mathbf{e}_j = \begin{pmatrix} \partial_s\psi & 0 \\ 0 & r\sin\psi \end{pmatrix}. \qquad (A24)$$

The flow of cortical material within the actomyosin layer and deformations of the cortex can be expressed in terms of a velocity flowfield $\mathbf{v}$ on the surface, which we decompose into its tangential and normal parts

$$\mathbf{v} = v^i\mathbf{e}_i + v^n\mathbf{n}, \qquad (A25)$$

such that $v^s$ corresponds to the velocity of cortical flows and $v^n$ is related to deformations of the surface. We assume $v^\phi = 0$ here. The symmetric velocity gradient tensor reads

$$v_{ij} = \frac{1}{2}\left(\mathbf{e}_i\cdot\partial_j\mathbf{v} + \mathbf{e}_j\cdot\partial_i\mathbf{v}\right) = \begin{pmatrix} \partial_s v^s + v^n\partial_s\psi & 0 \\ 0 & r(v^s\cos\psi + v^n\sin\psi) \end{pmatrix}. \qquad (A26)$$

Next, we discuss the form of the tension induced by the alignment of actin filaments with the cortical layer. We write the following expression:

$$t^i_{j,\text{nematic}} = f_1(\mathbf{Q}^2)\,\delta^i_j + f_2(\mathbf{Q}^2)Q^i_j, \tag{A27}$$

with $f_1$ and $f_2$ two functions of $\mathbf{Q}^2 = Q^i_j Q_i^j$, and $f_1(0) = 0$, as an isotropic stress component independent of $\mathbf{Q}$ has already been included in (**Equation A16**).

We assume here that in regions of the *C. elegans* embryo where $Q^\phi_\phi > 0$,

$$t^s_{s,\text{nematic}} \simeq 0, \tag{A28}$$

$$t^\phi_{\phi,\text{nematic}} = \xi Q, \tag{A29}$$

with $Q = Q^\phi_\phi = -Q^s_s > 0$, such that filament alignment orthogonal to the long axis of the embryo generates contractile stresses along the axis of filament alignment, but has negligible effect on stresses acting along the long axis of the embryo. To verify that **Equation (A28–A29)** are compatible with the tensorial symmetries imposed by (**Equation A27**), we note that by choosing $f_1$ and $f_2$ with the form

$$f_1(\mathbf{Q}^2) = \frac{\xi}{2\sqrt{2}}\sqrt{\mathbf{Q}^2},\ f_2(\mathbf{Q}^2) = \frac{\xi}{2}, \tag{A30}$$

one recovers Equation (A28S27) and (A29S28) for a diagonal tensor $\mathbf{Q}$ with $Q^\phi_\phi > 0$. This choice corresponds to the physical limit where filament alignment results in a higher contractile stress in the direction of filament alignment, while leaving stresses orthogonal to filament alignment unchanged. Such a choice appears compatible with experimental observations of polarization flows in the *C. elegans* embryo, where flows along the long axis of the embryo do not seem to be significantly influenced by filament alignment orthogonal to the direction of flows (**Mayer et al., 2010**).

With these definitions, we obtain the equations governing the flow in the cortical layer, and the shape of the cell, by combining the constitutive relation (**Equation A16**) with the force balance equations **Equation (A17–A18)**. We seek for a stationary solution to the cortex shape equation, and thus consider the case where $v^n = 0$. From the normal force balance **Equation (A17)**, we obtain the shape equation in a region where $Q > 0$:

$$P_{\text{int}} - P_{\text{ext}} = \partial_s \psi \left(\zeta + \left(\eta_b + \tfrac{1}{2}\eta_s\right)\partial_s v^s\right) + \frac{\sin\psi}{r}\left(\xi Q + \zeta + \left(\eta_b + \tfrac{1}{2}\eta_s\right)\frac{v^s\cos\psi}{r}\right) \\ + \left(\eta_b - \tfrac{1}{2}\eta_s\right)\left(\frac{\sin\psi}{r}\partial_s v^s + \frac{\cos\psi}{r}\partial_s\psi v^s\right). \tag{A31}$$

This equation relates the shape of the cell to the measured profiles of the cortex velocity field, the measured nematic order parameter, and the myosin dependent active isotropic and anisotropic stresses. We assume that the spatial profile of $\zeta$ depends linearly on the fluorescence intensity of the molecular motors in the cortex (i.e. NMY-2::GFP labeling) according to

$$\zeta = \zeta_0 \frac{I(s)}{\bar{I}}, \tag{A32}$$

where $I(s)$ is the profile of relative fluorescence intensity of cortical myosin, and $\bar{I}$ is the average intensity. To obtain an expression for the level of ingression during furrow initiation, we consider a reference state with a uniform distribution of myosin motors in the cortex, and consider deviations from this reference state

eLIFE Research article                    Cell Biology | Biophysics and Structural Biology

$$\zeta = \zeta_0 + \delta\zeta, \tag{A33}$$

with $\delta\zeta = \frac{\zeta_0(\mathrm{I}(s) - \bar{\mathrm{I}})}{\bar{\mathrm{I}}}$. In the reference state, the cortex does not form an ingression, the flow in the cortex is zero and the actin network is isotropic. Indeed, with $\mathrm{R}$ denoting the radius of the eggshell, and assuming it to be constant in $s$ over the relevant length scale, *Equations (A13) and (A31)* are solved by $\mathrm{r} = \mathrm{R}, \psi = \frac{\pi}{2}, \mathrm{v_s} = 0, \mathrm{Q^i_j} = 0, \ \mathrm{P_{int}} - \mathrm{P_{ext}} = \frac{\zeta_0}{\mathrm{R}}$. Performing a linear expansion around this state ($\mathrm{r} = \mathrm{R} + \delta\mathrm{r}, \psi = \frac{\pi}{2} - \partial_s\delta\mathrm{r}, \mathrm{v^s} = \delta\mathrm{v^s}, \mathrm{Q^i_j} = \delta\mathrm{Q^i_j}, \ \mathrm{P_{int}} - \mathrm{P_{ext}} = \frac{\zeta_0}{\mathrm{R}} + \delta\mathrm{P}$), we obtain the linearized shape equation for the level of ingression $\delta\mathrm{r}$

$$0 = \frac{\delta\zeta - \mathrm{R}\,\delta\mathrm{P}}{\zeta_0} + \frac{\xi\,\delta\mathrm{Q}^\phi{}_\phi}{\zeta_0} + \frac{2\eta_\mathrm{b} - \eta_\mathrm{s}}{2\zeta_0}\partial_s\delta\mathrm{v^s} - \frac{1}{\mathrm{R}}\delta\mathrm{r} - \mathrm{R}\,\partial_s^2\delta\mathrm{r}. \tag{A34}$$

The solution to this equation describes the deviation of the cell radius from the eggshell when the zygote forms an ingression at the onset of pseudocleavage and cytokinesis, in response to the isotropic and anisotropic tensions arising from spatial heterogeneities in the distribution of active motors. In the equation above, we have ignored for simplicity non-linear terms in the deviations from the reference state, although variations of myosin intensity around the mean are of order ~1; we do not expect however these additional terms to change our results qualitatively. The solution can be written as a function of two integration constants and three unknown parameters

$$\delta\mathrm{r} = \cos\!\left(\tfrac{s}{\mathrm{R}}\right)\mathrm{C}_1 + \sin\!\left(\tfrac{s}{\mathrm{R}}\right)\mathrm{C}_2 - \mathrm{R}\mathrm{p}_1 + \mathrm{p}_2\int_{s_0}^{s}\mathrm{d}s'\left[\cos\!\left(\tfrac{s-s'}{\mathrm{R}}\right)\delta\mathrm{v^s}(s')\right]$$
$$+\mathrm{p}_3\int_{s_0}^{s}\mathrm{d}s'\left[\sin\!\left(\tfrac{s-s'}{\mathrm{R}}\right)\delta\mathrm{Q}^\phi{}_\phi(s')\right] + \int_{s_0}^{s}\mathrm{d}s'\left[\sin\!\left(\tfrac{s-s'}{\mathrm{R}}\right)\left(\tfrac{\mathrm{I}(s')}{\bar{\mathrm{I}}} - 1\right)\right], \tag{A35}$$

with

$$\mathrm{p}_1 = \frac{\mathrm{R}\delta\mathrm{P}}{\zeta_0}, \mathrm{p}_2 = \frac{2\eta_\mathrm{b} - \eta_\mathrm{s}}{2\mathrm{R}\,\zeta_0}, \mathrm{p}_3 = \frac{\xi}{\zeta_0}. \tag{A36}$$

Given a set of boundary conditions, we can thus express the shape of the zygote as a function of its mechanical and geometric parameters, and the measured profiles of cortical flow and nematic order. At the points $s_1$ and $s_2$, the cortex of the embryo is touching the eggshell: the shape radius in these points is given by $\mathrm{R}$, and we assume is smoothly connected to the contact regions. Thus, $\delta\mathrm{r}$ fulfills the following four conditions

$$\delta\mathrm{r}(s = s_1) = 0, \partial_s\delta\mathrm{r}(s = s_1) = 0, \delta\mathrm{r}(s = s_2) = 0, \partial_s\delta\mathrm{r}(s = s_2) = 0. \tag{A37}$$

We determine the position of $s_1$ and $s_2$ from experimental measurements. We take into account that the cortex and eggshell have a soft contact due to elastic structures in between, such as the vitelline membrane. In order to obtain the position of contact points from experimental embryo shapes, we choose a threshold based on the estimated first derivative of the shape (the magnitude of the derivative is set to be larger than $\mathrm{h} = 0.04$). The boundary *Equations (A37)* are then used to determine the integration constants $\mathrm{C}_1$, $\mathrm{C}_2$ as well as the dimensionless pressure parameter $\mathrm{p}_1$ and the viscosity parameter $\mathrm{p}_2$. We interpolated the experimentally measured profiles of $\mathrm{Q}^\phi{}_\phi$, $\mathrm{I}(s)$, and $\mathrm{v^s}$ (see Appendix) with third-order polynomials and used these profiles to express the theoretical level of ingression of the embryo from *Equation (A35)*, as a function of a single unknown parameter, $\mathrm{p}_3 = \frac{\xi}{\zeta_0}$. Note that to enable the comparison to the experimentally determined ingression profile at pseudocleavage, we extrapolated the profile of nematic order at the anterior side on a length of about ~1 $\mu$m. The value of $\mathrm{p}_3$ is then adjusted by performing a fit of the predicted shape solution to the experimentally estimated values for the level of ingression during

Reymann *et al*. eLife 2016;5:e17807. DOI: 10.7554/eLife.17807                    24 of 25

pseudocleavage and cytokinesis. We have two datasets of size $N_{PC} = 22$, and $N_{CK} = 14$ for pseudocleavage and cytokinesis phases respectively, and minimize the difference between the data points and the theoretical curve using the objective function

$$S(p_3) = \frac{1}{N_{PC}} \sum_{i=1}^{N_{PC}} \left( \delta\hat{r}_{i,PC} - \delta r_{PC}(s_i; p_3) \right)^2 + \frac{1}{N_{CK}} \sum_{i=1}^{N_{CK}} \left( \delta\hat{r}_{i,CK} - \delta r_{CK}(s_i; p_3) \right)^2, \qquad \text{(A38)}$$

where PC and CK are short for pseudocleavage and cytokinesis respectively, and the hat denotes measured data. The parameter estimates are summarized in the table below. By fitting a single parameter to the data from both developmental phases, we achieve a good agreement between the theoretical description and the measurements (see *Figure 3e* and *Figure 3—figure supplement 3*, a constant offset was added to the theoretical ingression to take into account the thickness of the extra-embryonic layer). We find that the cortex deformations observed during furrow initiation are best matched with a myosin coupling ratio $p_3 = \frac{\xi}{\zeta_0} = 21 \pm 5$. Multiplying this ratio with the maximum order parameter along the embryo gives an estimate for the relative weight of anisotropic versus isotropic myosin-dependent stresses in driving the ingression formation. We find this ratio to be about 2 for pseudocleavage, and 3 for cytokinesis, indicating that anisotropic cortical tension due to actin filament alignment is crucial to explain the observed cell shapes during furrow initiation. Indeed, setting the stress-alignment coupling to zero ($p_3 = 0$) produces very poor agreement with the measured data (see *Figure 3e* and *Figure 3—figure supplement 3*).

## Parameters in the theoretical description

| Parameters | Pseudocleavage | Cytokinesis |
|---|---|---|
| Eggshell radius | | $14. \pm 0.1$ $\mu m$ |
| Pressure parameter $p_1$ | $1.4 \pm 0.4$ | $1.8 \pm 0.5$ |
| Viscosity parameter $p_2$ | $0.07 \pm 0.07$ $min/\mu m$ | $0.08 \pm 0.03$ $min/\mu m$ |
| Nematic to myosin ratio $p_3$ | | $21 \pm 5$ |

