## [Decision Letter]

[Editors’ note: a previous version of this study was rejected after peer review, but the authors submitted for reconsideration. The first decision letter after peer review is shown below.]

Thank you for submitting your work entitled "Cortical Flow Aligns Actin Filaments to Form a Cytokinetic Ingression" for consideration by *eLife*. Your article has been favorably evaluated by Naama Barkai (Senior Editor) and three reviewers, one of whom, Pekka Lappalainen, is a member of our Board of Reviewing Editors. Our decision has been reached after consultation between the reviewers. Based on these discussions and the individual reviews below, we regret to inform you that your work will not be considered further for publication in *eLife*.

All three reviewers found the topic of the study highly interesting, and stated that combining experiments and theory is certainly valuable for understanding the mechanisms underlying the formation of pseudocleavage furrow during interphase and a cytokinetic furrow during cell division.

However, as you will see from their comments below, the reviewers felt that the main conclusions of the study are not strongly supported by the experimental data presented in the manuscript. Especially reviewer #2 listed a number of weaknesses in the experimental design and interpretations of the data from RNAi and cell biological analyses. This, together with a lack of considering alternative theories, weakens the overall conclusions of the study. Therefore, an extensive amount of additional work (such as additional, more specific RNAi experiments and studying the possible contribution of a local zone of RhoA activation in this process) would be required to convincingly demonstrate that aligning actin filaments through cortical flow indeed drives the formation of contractile ring.

*Reviewer #1:*

The mechanism by which contractile actomyosin ring is formed for cytokinesis in metazoan cells is not fully understood, and for example whether the ring is generated through converging flows at the equatorial region or through a 'search-and-capture' mechanism is unknown. Here, Reymann et al. approached this question by studying the one cell embryo of *C. elegans*, where two constricting ingressions, a pseudocleavage furrow during interphase and cytokinetic furrow during cell division, are formed. Studies using experiments and theory provide evidence that aligning actin filament bundles through cortical flow drives the formation of ingression during both pseudocleavage and cytokinesis.

This work provides interesting new information concerning the mechanisms of cytokinesis in metazoan cells, and majority of the experiments are of good technical quality. However, there are some points that should be addressed to further strengthen the study. Furthermore, this manuscript was not particularly reader friendly and thus the authors should rewrite the text and figure legends to make the manuscript more accessible for a general reader.

1) My biggest concern is the rationale of the RNAi studies presented in Figure 4. The authors conclude form the phenotypes of *nop1, ani1* and *mlc-4* knockdown cells that decreased compression rates (but not the modification of the actomyosin gel organization) are responsible for the lack of pseudocleavage in *nop1* RNAi cells. However, because NOP-1 is an activator of RhoA, and is thus likely to affect myosin II activity through myosin light chain phosphorylation, both *nop1* and *mlc-4* depletions should have a similar effect on myosin II activity and organization of the 'acomyosin gel', which does not seem to be the case. This should be examined/clarified by additional experiments or the rationale of the experiment should be more clearly explained in the text.

2) The main difference between *nop1* and *mlc-*4 RNAi cells (based on Figure 4) is the lack of intense actin filament bundles in the absence of NOP-1. Because NOP-1 is an activator of RhoA it is likely that, in addition to defects in myosin II light chain phosphorylation, the dynamics of actin filaments in nop1 RNAi cells are more rapid as compared to control or *mlc-4* RNAi cells (due to effects of RhoA on cofilin activity). Would it thus be possible that the lack of proper actin bundles and myosin foci in *nop1* RNAi cells is due to rapid actin filament turnover and consequent defects in actin filament cross-linking?

3) From Figure 1 and Video 2 it seems that myosin II forms much larger foci before pseudocleavage furrow formation than before the cleavage furrow formation. Is this indeed the case in all cells? What is the reason for this, and can this phenomenon contribute to differences between these two ingressions?

*Reviewer #2:*

This manuscript combines experimental and computational approaches to investigate the mechanism of cleavage furrow ingression. In particular it seeks to evaluate the contribution that cortical flow based alignment contributes to cleavage furrow ingression. The authors conclude that filament alignment through flow is a highly generalized mechanism for cortical reorganization (Abstract) that is sufficient to account for furrow ingression. This strong conclusion is not well supported experimentally.

The authors examine whether cortical flow driven alignment or whether the SCPR model developed in *S. pombe* is responsible for cytokinetic ingression. However, they do not examine whether these are the only two possible mechanisms nor do they provide a rationale for focusing exclusively on these two mechanisms. Notably, they do not examine whether local RhoA activation could provide an alternative mechanism, despite the fact that there is copious evidence that RhoA can be activated locally. This local activation could provide an additional mechanism, indeed it appears to be the evolutionarily conserved mechanism in metazoa.

Overall, the manuscript is based predominantly on theory that is substantiated – in part – by experimental data. There are several major weaknesses. First, the theory is not described in a manner that is accessible to scientists outside of the biophysical modeling field. Second, the experimental perturbations are not exhaustive nor diagnostic. Thus, at least for a cell biological audience, the basis for the strong conclusions are not clearly articulated. Third, the authors fail to properly account for the knowledge concerning the perturbations used (see points 2, 3 below) thus weakening the overall conclusions. Fourth, the authors do not consider all reasonable alternative theories. To reiterate, the authors appear explore whether cortical flow induced filaments explain all furrowing behavior in the *C. elegans* embryo. There is extensive prior evidence that cortical flows exist in the early embryo (8509454, 15363415) and previous evidence demonstrating that these flows can contribute to filament alignment (8509454, 22918944). Thus it is not a novel insight that cortical flows can contribute to alignment and furrowing. However, the stronger statement, that furrowing is inextricably linked to cortical flows – which would be novel – is not convincingly tested.

Due to the role of myosin in promoting cell polarization, the *C. elegans* embryo cortex has extensive networks of cortical myosin during cytokinesis. For example, NOP-1 directed RhoA activation induces abundant cortical myosin during anaphase (as well as during polarization). In addition, CDC-42-dependent myosin accumulation results in cortical myosin that persists into anaphase and temporally overlaps with cytokinesis (PMID 19923324). However, furrows can form in the absence of such pools of actin and myosin. For example, in embryos with posteriorly localized spindles a posterior furrow forms in the absence of a broad zone of actomyosin (the cortical actomyosin is largely concentrated in the anterior). Moreover, local, centralspindlin-dependent induction of RhoA (PMID 22918944) can generate furrow directed flows of these populations of actomyosin. Thus, the authors have limited their analysis to situations with extensive cortical myosin and that would be expected to flow towards a local zone of RhoA activation. However, the authors do not explore the contribution that a local zone of RhoA activation would make to furrowing. If embryos do furrow in the absence of such flows, then the conclusion that flows are critical for furrow formation would be incorrect.

*Reviewer #3:*

Actomyosin cortex is reorganized during cytokinesis to form a contractile cytokinetic furrow. Using a combination of high-end imaging and modeling, Reymann et al. explore cortical actomyosin dynamics during furrow formation in a single-celled *C. elegans* embryo. They make use of two consecutive events occurring post-fertilization, pseudocleavage and cytokinesis proper. The authors propose that active cortical flows align actin filaments to help form cortical ingression. This work is related to a previous theoretical study by one of the co-authors that suggested that the flow could be sufficient to reorient actin filaments tangentially. The authors further propose that material properties of the actomyosin cortex are different during pseudocleavage and cytokinesis. The imaging data is generally of high quality and the theory is interesting. The story is really about pseudocleavage rather than cytokinesis but I don't think that this makes it potentially less interesting. With the caveat that actin biophysics is not my area of research I do not have issues with the central premise of the work. Having said that, I feel that the authors occasionally hype-up the novelty of the story and also sometimes over-interpret their results. I will give a few examples below. It is possible that some of the issues could be addressed with re-writing, but others need additional experiments.

From the graphs and images shown in Figure 4, it looks like flow velocity, compression rate and even the overall architecture of the cortical actomyosin are comparable in *nop-1* and *ani-1* embryos. Yet filaments align in *ani-1* cells. Why is that? I wonder if probing cortical tension with laser ablation might give insights into differences in cortical properties between these conditions.

Related to the previous point, it is now clear why the authors summarize their data differently in Figure 4 for the two RNAi conditions – e.g. why polarizing flow is represented as normal in anillin mutant but defective in *nop-1*? The graphs in Figure 4 show similar values. Similarly, why pseudocleavage ingression is shown as partially flawed in *ani-1*, while there is nothing obviously wrong with ingression in Figure 4? This entire panel in Figure 4 needs a wild type control to make comparisons easier.

It would be useful if the authors extended their analyses to other abnormal situations, in particular looking into conditions that directly affect actin filaments rather than myosin function (e.g. actin RNAi embryos that fail pseudocleavage but not cytokinesis, or cofilin).

It is not clear to me why the proposed model does not consider active filament alignment through motor activity. There is indeed no myosin enrichment at the site of furrowing in pseudocleavage but the myosin is still present there and presumably its activity could be regulated.

The cortical actomyosin is clearly different in *nop-1* or *ani-1* embryos – but what does this more homogeneous network mean in terms of cortical properties? Is tension more evenly distributed?

Do filaments align during initial cortical contractions prior to pseudocleavage?

Now, a few examples of what I mean by somewhat misleading statements:

The Abstract states that it is "unknown" how cortex "reorganizes itself to generate the contractile ring" but surely there is a lot of knowledge in the field about this process.

In Introduction, the authors posit that two distinct mechanisms can explain metazoan ring formation and propose to sort out which one is correct. Yet, one of these hypotheses was suggested for fission yeast, and even then it remains a hypothesis, with other groups demonstrating that rings can be assembled in the absence of cortical nodes required for search-and-capture mechanism to function. On the other hand, there is a lot of evidence of converging cortical flows during cytokinesis and actin filament alignment in metazoan furrows.

Related to Figure 4, the title is misleading – anillin is not the only bundler and so the authors can't say "Actin bundling is dispensable for actin alignment". Related to Figure 4, perhaps the authors could think of a better heading for *nop-1* chapter – that fact that *nop-1* is required for pseudocleavage has been known for 20 years.

[Editors’ note: what now follows is the decision letter after the authors submitted for further consideration.]

Thank you for submitting your article "Cortical flow aligns actin filaments to form a furrow" for consideration by *eLife*. Your article has been favorably evaluated by Naama Barkai (Senior Editor) and three reviewers, one of whom, Pekka Lappalainen (Reviewer #1), is a member of our Board of Reviewing Editors. Because two of the original reviewers declined to review the new submission, your manuscript was evaluated by the Reviewing Editor and two new external reviewers.

The reviewers have discussed the reviews with one another and the Reviewing Editor has drafted this decision to help you prepare a revised submission.

Summary:

Through the decades of research, many different models have been proposed to explain the formation of contractile forces at the cell division site. Here, Reymann et al. approached this question by studying the pseudocleavage furrow and cytokinetic furrow formation in the one cell embryo of *C. elegans*. This work provides evidence that cortical actin flows converging on the cell equator may compress and mechanically align the filaments to drive furrow formation. The authors developed an actin marker that allows imaging of actin bundles at unprecedented resolution, and applied sophisticated image analysis methods combined with theoretical model to test the contributions of possible mechanisms. Experimental data are well described by the theoretical model, and the predictions of the model were tested by mutants affecting flow patterns.

In general, this manuscript makes an important advance into our understanding of cytokinesis, and provides strong evidence that actin flows are critical in aligning the filaments at the division plane. However, there are few issues that should be thoroughly addressed to further strengthen the study.

Essential revisions:

1) The images of cortical actin filaments are striking, and were taken by a new LifeAct-Kate marker for F-actin. There is, however, some concern that the LifeAct probe may be stabilizing filaments/bundles, which can occur upon high levels of expression (Courtemanche et al., 2016 NCB); in the worst case scenario, the over-stabilization of filaments could bias the formation of the ring from pre-assembled filament bundles. The authors state in the methods that "overall cortical organization and flow dynamics appeared to not be affected by this reagent, foci lifetime, spacing, cortical flow velocities were similar to previous measurement with other fluorescent lines." These data should be shown as 'Supplemental information'. What are these other fluorescent markers? Assays that examine the dynamic turnover of filaments should be included. The dynamic behavior of actin should not only be compared with other LifeAct strains, but with other markers that do not directly affect actin dynamics.

2) In Figure 1, the authors used an 'active RhoA biosensor' to examine RhoA activity during pseudocleavage and cytokinesis. Which RhoA biosensor was used here? The details of this experiment should be described in 'Results' and 'Methods' sections. Furthermore, the active RhoA and myosin II appear to display very similar punctuate distributions with each other during pseudocleavage (Figure 1). The authors should thus examine whether the active RhoA and myosin II co-localize in these cells. Furthermore, they should confirm that the 'active RhoA biosensor' does not just simply detect phosphorylated myosin II in these cells.

3) One fairly important point that should be discussed in more detail, is connected to the method used to extract the nematic order parameter of the actin filaments within the cortex. Are the authors sure that they are measuring meaningful order parameters? It seems to me that once the filament concentration becomes too high (the filaments are then closer the optical resolution) the anisotropy in the local Fourier transform will start to wash away, leading to measurements that do not indicate the true orientational order within the cellular cortex. Because of the quantitative nature of the study, it is essential that their measurements of nematic order are quantitative. Therefore, this method needs to be justified, perhaps by applying it to a simpler system with a known order parameter.

[Editors' note: further revisions were requested prior to acceptance, as described below.]

Thank you for resubmitting your work entitled "Cortical flow aligns actin filaments to form a furrow" for further consideration at *eLife*. Your article has been favorably evaluated by Naama Barkai (Senior Editor) and three reviewers, one of whom is a member of our Board of Reviewing Editors..

The manuscript has been improved but there are some minor issues concerning one supplemental figure (see comments by reviewer #1 below) that need to be addressed before acceptance.

*Reviewer #1:*

The authors have now satisfactorily addressed the main concerns. However, the new data presented in Figure 2—figure supplement 5 should be properly discussed in the 'Results' and the constructs/strains used in these experiments (e.g. the worms expressing the PLST-1:GFP) should be described in the 'Methods' section.

*Reviewer #2:*

The reviewers have done a very good job addressing the major concerns on the manuscript. As described in their letter, they have added important data to the manuscript and clarifications in the text. I think that the manuscript is now acceptable for publication.

*Reviewer #3:*

The authors have thoroughly addressed all my concerns regarding the method used to measure the nematic order parameter. I believe that this interesting manuscript is suitable for publication in its current form.

---

## [Author Response]

[Editors’ note: the author responses to the first round of peer review follow.]

*All three reviewers found the topic of the study highly interesting, and stated that combining experiments and theory is certainly valuable for understanding the mechanisms underlying the formation of pseudocleavage furrow during interphase and a cytokinetic furrow during cell division.*

*However, as you will see from their comments below, the reviewers felt that the main conclusions of the study are not strongly supported by the experimental data presented in the manuscript. Especially reviewer #2 listed a number of weaknesses in the experimental design and interpretations of the data from RNAi and cell biological analyses. This, together with a lack of considering alternative theories, weakens the overall conclusions of the study. Therefore, an extensive amount of additional work (such as additional, more specific RNAi experiments and studying the possible contribution of a local zone of RhoA activation in this process) would be required to convincingly demonstrate that aligning actin filaments through cortical flow indeed drives the formation of contractile ring.*

The reviewer’s comments made us realize that we had pitched our work wrong, and that our manuscript lacked clarity in order to convey effectively our main findings. We have taken some time to do additional experiments and analysis, and modify and rewrite extensively our manuscript. First, as suggested we now consider an alternative active alignment theory(Search and Capture) in our analysis and can now show that this alternative mechanism does not contribute significantly to the equatorial alignment of filaments in our system. Second, we now investigate the contribution of a local zone of RhoA activationto this process. This was a great suggestion by the referees, and we now utilized an active RhoA probe to show that, for pseudocleavage there is in fact no local zone of RhoA activation at the cell equator, and filament alignment here happens only by flow. This is a major addition to our manuscript.

These two additions have very much shifted the focus of our work. Indeed, our first version was too concerned with detail and with the precise mechanism of function of various genes to the process, which we had tested by RNAi. This was correctly criticized by the referees, we had over-interpreted our results. In our overhaul of our manuscript and given the shift of focus, we have now substantially shortened this part of the manuscript and reduced it to the essential. Importantly, we remain descriptive here and refrain from drawing several of the conclusions that were criticized by the referees.

We think that much of the criticism that has arisen was due to our inability to provide a clear and central message in our original work, leaving the referees in a bit of a void with respect to what we are really showing. We have shortened the manuscript and condensed its message, clarifying that this is really a manuscript that seeks to investigate the physical mechanismsof furrow formation, and the biophysical aspects of cortical mechanicsthat ensure cytokinesis to take place. We test alternative models, and focus our analysis of the regulatory aspects more on RhoA than on the other genes. This work, however, does bridge into the biophysics and theory much more than many other papers do, and we think that this is also its core strength.

*Reviewer #1:*

*This work provides interesting new information concerning the mechanisms of cytokinesis in metazoan cells, and majority of the experiments are of good technical quality. However, there are some points that should be addressed to further strengthen the study. Furthermore, this manuscript was not particularly reader friendly and thus the authors should rewrite the text and figure legends to make the manuscript more accessible for a general reader.*

*1) My biggest concern is the rationale of the RNAi studies presented in Figure 4. The authors conclude form the phenotypes of nop1, ani1 and mlc-4 knockdown cells that decreased compression rates (but not the modification of the actomyosin gel organization) are responsible for the lack of pseudocleavage in nop1 RNAi cells. However, because NOP-1 is an activator of RhoA, and is thus likely to affect myosin II activity through myosin light chain phosphorylation, both nop1 and mlc-4 depletions should have a similar effect on myosin II activity and organization of the 'acomyosin gel', which does not seem to be the case. This should be examined/clarified by additional experiments or the rationale of the experiment should be more clearly explained in the text.*

This criticism, along with other criticism below on the RNAi studies presented, forced us to reconsider the main message of the work and the pitch of the entire paper. Thus in response to this comment we have significantly shortened the paragraph entitled “*nop-1 RNAi* abolishes pseudocleavage formation”, as well as the following paragraphs on RNAi experiments. We focus our discussion and conclusion on the core result, that reducing flow speeds results in reduced alignment of actin filaments.

NOP-1 indeed acts upstream of RhoA, and thus affects MLC-4, thereby reducing myosin activity. As a consequence, *nop-1* and *mlc-4 RNAi* both lead to decreased flow rates due to lower levels of active myosin, which we show, results in reduced local alignment and furrowing. However, NOP-1 also directly affects actin organization by affecting the activity of several other actin binding proteins (through anillin, formin or cofilin for example – as also pointed by the reviewer in the text concern), which would explain the difference of phenotypes between *nop-1* and *mlc-4 RNAi (mlc-4* affecting only myosin II activity). Importantly we here used mild *mlc-4 RNAi* conditions, instead of full knockdowns, in order to decrease progressively compressive flows without affecting the overall cortical organization (keeping the presence of myosin foci for example as shown in Figure 4—figure supplement 1). To answer the reviewers specific concern, *nop-1 RNAi* can have similar effects on myosin II activity than *mlc-4 RNAi* however the organization of the 'actomyosin gel' is clearly different in the two cases.

In response to this comment we have rewritten the corresponding section of the manuscript, condensed the corresponding figure, and focused our analysis and discussion on the only thing we can really conclude from the RNAi experiments, which is that reducing flow speeds results in reduced alignment of actin filaments.

*2) The main difference between nop1 and mlc-4 RNAi cells (based on Figure 4) is the lack of intense actin filament bundles in the absence of NOP-1. Because NOP-1 is an activator of RhoA it is likely that, in addition to defects in myosin II light chain phosphorylation, the dynamics of actin filaments in nop1 RNAi cells are more rapid as compared to control or mlc-4 RNAi cells (due to effects of RhoA on cofilin activity). Would it thus be possible that the lack of proper actin bundles and myosin foci in nop1 RNAi cells is due to rapid actin filament turnover and consequent defects in actin filament cross-linking?*

This is an interesting point, RhoA was shown to inhibit cofilin and cofilin increases turnover, so it is true that potentially *nop-1 RNAi* could increase turnover of actin structures. However, in light of the major overhaul of our work and the change of focus to a more biophysics-oriented manuscript, we think that this is beyond our main message now and have chosen not to discuss this issue in the main text.

In response to this comment we have rewritten the corresponding section of the manuscript, condensed the corresponding figure, and focused our discussion of the *mlc- 4* and *nop-1 RNAi* phenotypes on highlighting that in both, reducing flow speeds results in reduced alignment of actin filaments.

*3) From Figure 1 and Video 2 it seems that myosin II forms much larger foci before pseudocleavage furrow formation than before the cleavage furrow formation. Is this indeed the case in all cells? What is the reason for this, and can this phenomenon contribute to differences between these two ingressions?*

This is a very good observation – indeed myosin seems to generate larger foci at cytokinesis. We currently have a manuscript under review that is concerned with the mechanisms of myosin pulsation in *C. elegans*, showing that there is an active RhoA spatiotemporal oscillator driving myosin foci formation in *C. elegans*. Importantly, modulation of active RhoA levels affects the spatiotemporal characteristics (size, timescale) of this pulsation. The observation from the referee thus suggests that the RhoA levels are different in pseudocleavage and cytokinesis – and they are, as we show with our new experiments using an active RhoA probe (Figure 1). However, while this would be really interesting to understand more, we do think that this is beyond our core message and we have chosen not to investigate this at greater detail in this manuscript.

*Reviewer #2:*

*This manuscript combines experimental and computational approaches to investigate the mechanism of cleavage furrow ingression. In particular it seeks to evaluate the contribution that cortical flow based alignment contributes to cleavage furrow ingression. The authors conclude that filament alignment through flow is a highly generalized mechanism for cortical reorganization (Abstract) that is sufficient to account for furrow ingression. This strong conclusion is not well supported experimentally.*

*The authors examine whether cortical flow driven alignment or whether the SCPR model developed in S. pombe is responsible for cytokinetic ingression. However, they do not examine whether these are the only two possible mechanisms nor do they provide a rationale for focusing exclusively on these two mechanisms. Notably, they do not examine whether local RhoA activation could provide an alternative mechanism, despite the fact that there is copious evidence that RhoA can be activated locally. This local activation could provide an additional mechanism, indeed it appears to be the evolutionarily conserved mechanism in metazoa.*

Our complete overhaul of the manuscript now takes into explicit consideration these very important and interesting comments, and we hope that the referee agrees that we have taken his/her concerns seriously and addressed them to the best way we can.

In response to the first concern here, from a physical perspective two different types of mechanism can explain an increase of actin filaments alignment for cytokinetic ingression, either an active alignment process such as by molecular motors in the Search and Capture model or a passive alignment due to mechanical constraints imposed by compressive flows. In this new version of the manuscript we now distinguish between these two alternative mechanical contributions. We conclude that compression by flow and not myosin-based active alignment is the driving force of ring formation in both pseudocleavage and cytokinesis (see Results and Discussion, first and fifth paragraphs and Figure 3—figure supplement 2).

In response to the second reviewer’s concern, we also now discuss the contribution of a local active RhoA equatorial zone during the formation and ingression of furrows and use a probe for active RhoA. This was a great suggestion by the referee, we find that this mechanism does not operate during pseudocleavage since active RhoA (like myosin itself) is not enriched at the position of the pseudocleavage ingression (see the new Figure 1). For this reason we can conclude that pseudocleavage cannot be triggered by local zone of RhoA activation (which is the basis of an active Search and Capture mechanism) but instead results from compression based alignment. Thus we conclude that pseudocleavage solely arises as a by-product of flow-driven cell polarization. This important suggestion has significantly improved our manuscript, and caused us to shift our focus more on pseudocleavage and the physical mechanisms of filament alignment, and importantly also prompted us to compare the contributions from flow based alignment and search-and capture.

*Overall, the manuscript is based predominantly on theory that is substantiated – in part – by experimental data. There are several major weaknesses. First, the theory is not described in a manner that is accessible to scientists outside of the biophysical modeling field. Second, the experimental perturbations are not exhaustive nor diagnostic. Thus, at least for a cell biological audience, the basis for the strong conclusions are not clearly articulated. Third, the authors fail to properly account for the knowledge concerning the perturbations used (see points 2, 3 below) thus weakening the overall conclusions. Fourth, the authors do not consider all reasonable alternative theories. To reiterate, the authors appear explore whether cortical flow induced filaments explain all furrowing behavior in the C. elegans embryo. There is extensive prior evidence that cortical flows exist in the early embryo (8509454, 15363415) and previous evidence demonstrating that these flows can contribute to filament alignment (8509454, 22918944).*

As mentioned earlier we realize that our previous manuscript lacked clarity in order to convey effectively the main findings of our work. We thus took some time to modify it extensively.

With respect to the first concern, we rephrased in simpler words our description of the biophysical theory presented here (the fourth paragraph of the Results and Discussion section detailing equation 1 and defining the parameters used). The schematic representations presented in the Figures (such as 2B, 3A or D) should also be a helpful aid for scientists outside of the biophysical modeling field.

With respect to the second and third concern, we agree that the analysis and discussion of the perturbation experiments in the first version of our manuscript was superficial and lacking substance. We have now performed additional experiments and importantly, have streamlined our conclusion, including only those required to substantiate the fact that compressive flows are required for the observed alignment process. It is not our intention in this paper to decipher the whole molecular requirement for cortical flows establishment, neither the individual molecular regulation of cortical material properties (we have another manuscript in preparation on this topic) and their impact on compression based alignment process. Finally, as mentioned above, our new findings have shifted the focus of the work this is why we have now substantially shortened this part of the manuscript.

With respect to the fourth concern and as mentioned above, we have taken this seriously and now consider an alternative physical theory; the Search-and-Capture model brought up by the referee in which direct active alignment by motors contributes to ring formation (see Results and Discussion, fifth paragraph and Figure 3—figure supplement 2). This was a great suggestion, which has helped us to improve our work. Importantly, we can now show that compression by flow and not myosin-based active alignment is the driving force of ring formation in both pseudocleavage and cytokinesis.

*Thus it is not a novel insight that cortical flows can contribute to alignment and furrowing. However, the stronger statement, that furrowing is inextricably linked to cortical flows – which would be novel – is not convincingly tested.*

In the 30 years following the seminal publication from White and Borisy, it is true that this mechanism of flow based filaments alignment was often mentioned and has been well accepted in the field. On one hand, observations from fixed cells have shown actin filaments alignment at the cell equator of dividing cells (Maupin and Pollard 1986, Fishkind and Wang 1993, Kamasaki et al. 2007), one the other hand flows towards the cell equator were observed in live cells. However, it has never been tested if this mechanism is actually at work. This requires a framework for testing the hypothesis. This framework was only recently provided by active gel theory, and we here for the first time show that compression and shear flow indeed aligns filaments. This is a novel and important addition to our understanding of actomyosin networks.

With regards to the second statement, it is clear that not all furrows are generated by cortical flow and compression. Hence, flow-based alignment is not the only mechanism and furrowing cannot be inextricably linked to cortical flows. Search and Capture is a prominent alternative mechanism, and we now show in our revised manuscript that it contributes via active alignment, albeit only to a small degree (Results and Discussion, fifth paragraph).

*Due to the role of myosin in promoting cell polarization, the C. elegans embryo cortex has extensive networks of cortical myosin during cytokinesis. For example, NOP-1 directed RhoA activation induces abundant cortical myosin during anaphase (as well as during polarization). In addition, CDC-42-dependent myosin accumulation results in cortical myosin that persists into anaphase and temporally overlaps with cytokinesis (PMID 19923324). However, furrows can form in the absence of such pools of actin and myosin. For example, in embryos with posteriorly localized spindles a posterior furrow forms in the absence of a broad zone of actomyosin (the cortical actomyosin is largely concentrated in the anterior). Moreover, local, centralspindlin-dependent induction of RhoA (PMID 22918944) can generate furrow directed flows of these populations of actomyosin. Thus, the authors have limited their analysis to situations with extensive cortical myosin and that would be expected to flow towards a local zone of RhoA activation. However, the authors do not explore the contribution that a local zone of RhoA activation would make to furrowing. If embryos do furrow in the absence of such flows, then the conclusion that flows are critical for furrow formation would be incorrect.*

The referee has prompted us to investigate active RhoA, and we now show that the pseudocleavage furrow arises without an equatorial zone of RhoA activation (new Figure 1). We can thus conclude that furrow formation is possible in the absence of a local zone of RhoA activation. In the revised manuscript we also included in our theory the contribution of direct myosin accumulation (such as the consequence of an equatorial local zone of RhoA activation) by the addition of an active alignment contribution of myosin molecular motors (equation 3). However, it is impossible for us to test experimentally a condition where a local zone of RhoA activation would be present in the total absence of flow, as RhoA by recruiting active myosin leads to a gradient of contractility which results in the generation of flows.

In response to the concerns raised here, we have rewritten much of the manuscript, focusing the work on pseudocleavage and the physical mechanisms that give rise to filament alignment.

*Reviewer #3:*

*Actomyosin cortex is reorganized during cytokinesis to form a contractile cytokinetic furrow. Using a combination of high-end imaging and modeling, Reymann et al. explore cortical actomyosin dynamics during furrow formation in a single-celled C. elegans embryo. They make use of two consecutive events occurring post-fertilization, pseudocleavage and cytokinesis proper. The authors propose that active cortical flows align actin filaments to help form cortical ingression. This work is related to a previous theoretical study by one of the co-authors that suggested that the flow could be sufficient to reorient actin filaments tangentially. The authors further propose that material properties of the actomyosin cortex are different during pseudocleavage and cytokinesis. The imaging data is generally of high quality and the theory is interesting. The story is really about pseudocleavage rather than cytokinesis but I don't think that this makes it potentially less interesting. With the caveat that actin biophysics is not my area of research I do not have issues with the central premise of the work. Having said that, I feel that the authors occasionally hype-up the novelty of the story and also sometimes over-interpret their results.*

The referee could have not formulated his criticism more clearly. In response to this and other comments we have now essentially rewritten the manuscript, and toned down our language in this regard.

*I will give a few examples below. It is possible that some of the issues could be addressed with re-writing, but others need additional experiments.*

*From the graphs and images shown in Figure 4, it looks like flow velocity, compression rate and even the overall architecture of the cortical actomyosin are comparable in nop-1 and ani-1 embryos. Yet filaments align in ani-1 cells. Why is that? I wonder if probing cortical tension with laser ablation might give insights into differences in cortical properties between these conditions.*

This is a very good observation, for easier comparison we now also added Figure 4—figure supplement 2. In ANI-1 depleted cells filaments interactions are less stable (decreased level of bundling agents) thus could be aligned more easily than in the wild type. NOP-1 acting upstream of several actin organizing mechanisms (one of which being ANI-1), it is unclear if the level of bundling agent would be similar in *nop-1* and *ani-1 RNAi* or if some additional mechanisms (such as through cofilin regulation for example) could be at work in NOP-1 depleted embryos. It is thus probable that the material cortical properties of *nop-1* and *ani-1 RNAi* embryos would be different; explaining the better ability to align filaments in ANI-1 depleted embryos compared to NOP-1 depleted ones. Regarding reviewer’s #3 suggestion, we considered the use of laser ablation but dismissed this option since laser ablation is a good tool to probe cortical tension and evaluate some material properties such as the Maxwell time, the hydrodynamic length, the ratio of active stress and per-area friction (Saha et al., Biophysical Journal, 2016) but it is not yet possible to use it to obtain nematic material properties we discuss here.

*Related to the previous point, it is now clear why the authors summarize their data differently in Figure 4 for the two RNAi conditions – e.g. why polarizing flow is represented as normal in anillin mutant but defective in nop-1? The graphs in Figure 4 show similar values. Similarly, why pseudocleavage ingression is shown as partially flawed in ani-1, while there is nothing obviously wrong with ingression in Figure 4? This entire panel in Figure 4 needs a wild type control to make comparisons easier.*

This was an error from our side, anillin depleted embryos should also have been summarized as defective in that table. Regarding its level of ingression and the outer level of ingression (outer ingression taking egg shell reference, without membrane overlap) seems very similar to that of the non-RNAi condition in anillin depleted embryos, however we did not observe any membrane overlap nor real pseudocleavage formation. Thus some cell shape change is observed, probably resulting of the observed alignment and/or change of

material properties but no proper cell furrowing is formed.

In response to this comment we have removed the table originally Figure 4 that over simplified the observed phenotypes, we have added Figure 4—figure supplement 2 to make comparisons easier in between all the different experimental conditions, and have substantially précised our observations in the Figure legends of Figure 4—figure supplement 1 and Figure 4—figure supplement 2.

*It would be useful if the authors extended their analyses to other abnormal situations, in particular looking into conditions that directly affect actin filaments rather than myosin function (e.g. actin RNAi embryos that fail pseudocleavage but not cytokinesis, or cofilin).*

As already pointed out by reviewer #1 and #2 the perturbation experiments presented here are not an exhaustive case study of all abnormal situations. We have performed many more RNAi (also targeting directly actin organization) in our laboratory. Actin organization, dynamics and cortical contractility are in fact closely related and under the control of numerous molecular players, all of them controlling the material properties of this gel like material. Understanding such molecular control in detail is a long process and currently one of the main interests of our laboratory. However, in the manuscript presented here we have decided to include only those required to substantiate the fact that compressive flows are required for the observed alignment process. It is not our intention in this paper to decipher the whole molecular requirement for cortical flows establishment, neither the individual molecular regulation of cortical material properties (manuscript in preparation). In fact, the perturbation of the actin cortex dynamics is intrinsically linked to its material properties and thus affects polarization flows, making us difficult to disentangle the impact on compressive flows on the alignment process. Finally, as mentioned above, our new findings have shifted the focus of the work this is why we have now substantially shortened this part of the manuscript and reduced it to the essential.

*It is not clear to me why the proposed model does not consider active filament alignment through motor activity. There is indeed no myosin enrichment at the site of furrowing in pseudocleavage but the myosin is still present there and presumably its activity could be regulated.*

Once again this is a very good point shared by all three reviewers. The revised manuscript now fully includes active filament alignment through motor activity (see Results and Discussion, fifth paragraph and Figure 3—figure supplement 2), which is in fact a Search and Capture mechanism. The addition of such an extra term in our theory enables us to estimate its relative contribution compared to compression-based alignment. However, “We conclude that compression by flow and not myosin-based active alignment is the driving force of ring formation in both pseudocleavage and cytokinesis.”

*The cortical actomyosin is clearly different in nop-1 or ani-1 embryos – but what does this more homogeneous network mean in terms of cortical properties? Is tension more evenly distributed?*

What we were trying to refer to is the actin filaments distribution – a more isotropic and uniform distribution as compared to an aster like bundled organization observed in the presence of myosin foci. Even though this is a very interesting question, in the scope of this paper we do not relate this observation to some physical cortical properties. Understanding such molecular control in detail is currently one of the main interests of our laboratory and a manuscript by Naganathan et al. is in fact in preparation on this topic.

In response to this and other comments we have reduced significantly our discussion of the RNAi phenotypes and only focus on those aspects that focus on flow and compression, hence we have not pursued this any further.

*Do filaments align during initial cortical contractions prior to pseudocleavage?*

Prior to pseudocleavage during the initial cortical contractions, we observe only a little bit of local alignment (actin rings at the periphery of blebs or aster like patterns around foci). However, these are only transient local alignment patterns without any preferred direction along the axis of the embryo, and no large-scale orientation pattern is observed prior flow.

*Now, a few examples of what I mean by somewhat misleading statements:*

*The Abstract states that it is "unknown" how cortex "reorganizes itself to generate the contractile ring" but surely there is a lot of knowledge in the field about this process.*

Fixed. We have removed this statement.

*In Introduction, the authors posit that two distinct mechanisms can explain metazoan ring formation and propose to sort out which one is correct. Yet, one of these hypotheses was suggested for fission yeast, and even then it remains a hypothesis, with other groups demonstrating that rings can be assembled in the absence of cortical nodes required for search-and-capture mechanism to function. On the other hand, there is a lot of evidence of converging cortical flows during cytokinesis and actin filament alignment in metazoan furrows.*

Abstract and Introduction were rewritten, we now compare the contributions from search- and-capture and flow-based alignment.

*Related to Figure 4, the title is misleading – anillin is not the only bundler and so the authors can't say "Actin bundling is dispensable for actin alignment". Related to Figure 4, perhaps the authors could think of a better heading for nop-1 chapter – that fact that nop-1 is required for pseudocleavage has been known for 20 years.*

Fixed. We no longer make this statement.

[Editors' note: the author responses to the re-review follow.]

*In general, this manuscript makes an important advance into our understanding of cytokinesis, and provides strong evidence that actin flows are critical in aligning the filaments at the division plane. However, there are few issues that should be thoroughly addressed to further strengthen the study.*

*Essential revisions:*

*1) The images of cortical actin filaments are striking, and were taken by a new LifeAct-Kate marker for F-actin. There is, however, some concern that the LifeAct probe may be stabilizing filaments/bundles, which can occur upon high levels of expression (Courtemanche et al., 2016 NCB); in the worst case scenario, the over-stabilization of filaments could bias the formation of the ring from pre-assembled filament bundles. The authors state in the methods that "overall cortical organization and flow dynamics appeared to not be affected by this reagent, foci lifetime, spacing, cortical flow velocities were similar to previous measurement with other fluorescent lines." These data should be shown as 'Supplemental information'. What are these other fluorescent markers? Assays that examine the dynamic turnover of filaments should be included. The dynamic behavior of actin should not only be compared with other LifeAct strains, but with other markers that do not directly affect actin dynamics.*

Indeed, Lifeact can affect actin dynamics if expressed at high levels. For our work it is important to verify that our LifeAct construct does not impact flow and filament alignment profiles. We had performed a verification as part of our experiments, but had not added this information to the original manuscript. In response to this comment, we have performed additional verifications and added all this data, the new verifications as well as our previous work, to the supplement as requested.

In particular, we had checked if anteroposterior flow-fields are similar with and without LifeAct. We found that anteroposterior flow-fields are similar in worms expressing NMY-2::GFP (Mayer et al) or NMY-2:mKate2 and our worms here expressing different constructs of LifeAct (Figure 2—figure supplement 5). Therefore, flow- and compression fields are unaffected by the presence of LifeAct in our strain. We’d also like to explicitly point out that the pseudocleavage furrow was discovered in unlabeled embryos, and can be seen in every embryo by use of DIC imaging.

Second to investigate if there is an impact on the alignment profile, we compared three different LifeAct constructs with distinct expression levels and different fluorophores. We find that both anteroposterior flowfields and the profile of nematic order are not significantly different between these three lines (Figure 2—figure supplement 5). Note that for Lifeact:GFP the peak in nematic order is slightly lower than for the other two lines, but this is likely due to our quantification method and the reduced LifeAct fluorescence intensity signal (see also below). The general shape, however, is well preserved among the three lines, all three strains showed similar nematic order profiles with a peak of vertical alignment at the cell equator during pseudocleavage. Taken together, this indicates that we are using LifeAct at concentrations that do not significantly impact actin and actomyosin cortex dynamics.

Finally, we verified that the dynamics of cytokinetic ring closure, another process that sensitively depends on actomyosin mechanics, appears similar both with and without LifeAct (see Figure 6).

Author response image 1.Actomyosin ring closure dynamics.**DOI:**
http://dx.doi.org/10.7554/eLife.17807.029

Note that Lifeact is the best probe we have to follow in real time the dynamics of the actin filaments in *C. elegans*. All our efforts to directly label actin have led to severe actomyosin phenotypes that prohibit any type of quantitative analysis.

Finally it is worthwhile noting that some actin bundling proteins are present in the cortex, such as plastin, and are stabilizing actin bundles in the cortical layer, especially in the cytokinetic ring as shown by worms expressing the PLST-1:GFP (endogenous CRISPR labeling, Figure 2—figure supplement 5). Thus the cell autonomously stabilizes aligned bundles of actin filaments and it is likely that doing so reinforces the process of flow alignment coupling.

*2) In Figure 1, the authors used an 'active RhoA biosensor' to examine RhoA activity during pseudocleavage and cytokinesis. Which RhoA biosensor was used here? The details of this experiment should be described in 'Results' and 'Methods' sections. Furthermore, the active RhoA and myosin II appear to display very similar punctuate distributions with each other during pseudocleavage (Figure 1). The authors should thus examine whether the active RhoA and myosin II co-localize in these cells. Furthermore, they should confirm that the 'active RhoA biosensor' does not just simply detect phosphorylated myosin II in these cells.*

We here used a RhoA biosensor developed by the Glotzer lab (GFP::AHPH, *C. elegans* strain MG617, Tse et al., MBoC 2012). This sensor consists of GFP fused to the C-terminal portion of *C. elegans* anillin, which contain its conserved region (AH) and pleckstrin homology (PH) domain. It lacks the N-terminal myosin and actin-binding domains but retains its RhoA-binding domain. This information is now more clearly detailed in the Materials and methods section of the manuscript (first paragraph).

It is correct that active RhoA and myosin II display similar and overlapping foci patterns during pseudocleavage. In fact, we have in another manuscript from our lab that is currently under review with *eLife* as well, quantified the spatial and temporal correlation between active Rho and myosin (Nishikawa et al., "Controlling contractile instabilities in the actomyosin cortex", e*Life*: 12-07-2016-ISRA-*eLife*-19595). In this other work we report that feedback between active Rho and myosin induces a contractile instability in the cortex and discover that an independent RhoA pacemaking oscillator controls this instability, generating a pulsatory pattern of myosin foci. In Figure 1 of this other manuscript, where a spatio-temporal correlation analysis reveals that active Rho and myosin oscillate together (1F) leading to the generation of pulsatile behavior within the cortex (1C). Importantly however, our analysis reveals at a quantitative level the kinetics of myosin recruitment by active RhoA (1H). This figure shows that increasing active RhoA levels leads to an increase of myosin (red line), while the levels of myosin have no impact on active RhoA levels (green line). Hence our biosensor reports on RhoA activity, and does not just simply detect phosphorylated myosin II.

*3) One fairly important point that should be discussed in more detail, is connected to the method used to extract the nematic order parameter of the actin filaments within the cortex. Are the authors sure that they are measuring meaningful order parameters? It seems to me that once the filament concentration becomes too high (the filaments are then closer the optical resolution) the anisotropy in the local Fourier transform will start to wash away, leading to measurements that do not indicate the true orientational order within the cellular cortex. Because of the quantitative nature of the study, it is essential that their measurements of nematic order are quantitative. Therefore, this method needs to be justified, perhaps by applying it to a simpler system with a known order parameter.*

The referee raises an important point here. Of course, we can only analyze what we can see in the light microscope. As such, our measured order parameter can only be viewed as an approximation of the actual nematic order parameter of the network. The method does capture general features of the state of alignment, which we tested by use of synthetically generated test images, and we have now added Figure 2—figure supplement 3 and 4 to demonstrate this. Here we generated random networks with an increasing amount of vertical orientation. Notably, our quantification method captures the anisotropy in these synthetic network, and measures higher nematic order parameters for networks with a higher degree of anisotropy.

As the referee correctly points out, however, this only works if there is a significant amount of contrast in the image. If filaments get too dense and too close and are separated by distances that are of the order of the resolution in the light microscope, we are no longer able to detect this. And this is unavoidable; this is the price we pay for pursuing a live cell analysis.

We are confident, however, that we do capture well the overall state of alignment. In particular we have chosen rather large box sizes for the Fourier analysis (3.15 x 3.15 µm), and our measures of nematic order represent averages over this length scale. Even the strongest of bundles do not reach a width that is near this value, hence the images that we fourier-transform have a sufficient amount of contrast and anisotropies are well detected throughout the cell. For this reason, we also analyze only the early stages of furrow formation, and stop before there is a pronounced ingressing ring with many closely positioned bundles that can no longer be individually resolved, and for which we would not be able to determine the state of alignment by use of our analysis.

Furthermore, nematic order should be independent of network density. We tested this by generating synthetic networks of increasing densities, for a range of densities that resemble densities we observe in the embryo. Figure 2—figure supplement 4 shows that indeed, the profile of nematic order in a synthetically generated anisotropic network is to first order independent of network density when we use our algorithm, consistent with this prediction.

The measurement of nematic order does, however, depend on the noise and on the overall contrast in the image. Again by using synthetic networks, we find that nematic order decreases with increasing noise (Figure 2—figure supplement 4). For this reason, it is important to use the mKate2 LifeAct strain that is optimized for *C. elegans* and particularly bright. Importantly, we used the same imaging conditions and the same strain for all our experiments (otherwise stated), with similar amounts of noise in the images and with a similar image contrast.

[Editors' note: further revisions were requested prior to acceptance, as described below.]

*The manuscript has been improved but there are some minor issues concerning one supplemental figure (see comments by reviewer #1 below) that need to be addressed before acceptance.*

*Reviewer #1:*

*The authors have now satisfactorily addressed the main concerns. However, the new data presented in Figure 2—figure supplement 5 should be properly discussed in the 'Results' and the constructs/strains used in these experiments (e.g. the worms expressing the PLST-1:GFP) should be described in the 'Methods' section.*

We have now addressed the remaining issue of reviewer #1 considering Figure 2—figure supplement 5, by adding a sentence to the Results section that discusses this figure appropriately. We also updated the Methods section that describes how these new strains were obtained.